**nature** **COMMUNICATIONS**

# Frontotemporal coordination predicts working memory performance and its local neural signatures

Ehsan Rezayat [1], Mohammad-Reza A. Dehaqani [1,2✉], Kelsey Clark[3], Zahra Bahmani[4], Tirin Moore [5,6] & Behrad Noudoost [3✉]

Neurons in some sensory areas reflect the content of working memory (WM) in their spiking activity. However, this spiking activity is seldom related to behavioral performance. We studied the responses of inferotemporal (IT) neurons, which exhibit object-selective activity, along with Frontal Eye Field (FEF) neurons, which exhibit spatially selective activity, during the delay period of an object WM task. Unlike the spiking activity and local field potentials (LFPs) within these areas, which were poor predictors of behavioral performance, the phase-locking of IT spikes and LFPs with the beta band of FEF LFPs robustly predicted successful WM maintenance. In addition, IT neurons exhibited greater object-selective persistent activity when their spikes were locked to the phase of FEF LFPs. These results reveal that the coordination between prefrontal and temporal cortex predicts the successful maintenance of visual information during WM.

[1] School of Cognitive Sciences, Institute for Research in Fundamental Sciences (IPM), Tehran, Iran. [2] Cognitive Systems Laboratory, Control and Intelligent Processing Center of Excellence (CIPCE), School of Electrical and Computer Engineering, College of Engineering, University of Tehran, Tehran, Iran. [3] Department of Ophthalmology and Visual Sciences, University of Utah, Salt Lake City, UT, USA. [4] Department of Biomedical Engineering, Tarbiat Modares University, Tehran, Iran. [5] Department of Neurobiology Stanford University, Stanford, CA, USA. [6] Howard Hughes Medical Institute, Stanford University, Stanford, CA, USA. ✉email: dehaqani@ut.ac.ir; behrad.noudoost@utah.edu

Iigh-level brain areas such as prefrontal cortex (PFC) and parietal areas are believed to be central in the control and execution of behavioral plans[1]. How these areas interact with sensory areas to maintain task-relevant information is not well understood. Neurons in prefrontal cortex exhibit sustained spiking activity that encodes the content of WM[2]. However, since feature selectivity in some PFC areas seems insufficient to match the resolution of WM, a sensory recruitment model of WM has been suggested, in which brain areas involved in sensory processing also contribute to memory maintenance[3,4]. Such models involve interactions between PFC and sensory areas during memory maintenance. Variations of this idea range from a more modular perspective, in which specific portions of PFC exhibiting memory activity send that signal to sensory areas[5,6], or a split of abstract vs. detailed information[7] (perhaps varying based on task demands[8]), to more distributed versions emphasizing the content-specific communication between areas rather than spiking activity within either[9]. These conceptual models are not necessarily mutually exclusive, but they all predict that memory-related activity in sensory areas should be related to WM performance, a prediction that is discrepant with much of the existing neural data. Studies have shown that even when the content of WM is present in spiking activity within sensory areas, it is either only weakly correlated[10,11] or not correlated at all[12,13] with behavioral performance, casting doubt on its role in memory maintenance.

In order to examine how the PFC interacts with sensory areas in support of WM maintenance, we simultaneously recorded spiking activity and LFPs within the FEF and IT in monkeys performing an object WM task. We sought to determine which aspects of the neural response were most closely linked to successful WM maintenance. We found that neither the spiking activity nor the LFP power spectrum within either of these areas was a strong predictor of the animal's performance on this task. However, the synchronization between the two areas (phase–phase locking, PPL), particularly in the beta band, was a key predictor of successful object memory maintenance. Even though the object selectivity within IT did not differ between correct and wrong trials, the timing of spikes in IT were coordinated with the phase of the FEF LFP, and when the two areas were more strongly locked, IT spiking activity showed greater selectivity for the object held in memory. Thus, beta band synchrony between IT and FEF predicted both behavioral performance and the strength of object-selective persistent activity in IT.

## Results

We recorded spiking activity and LFP signals simultaneously from FEF and IT sites with overlapping receptive fields (RFs) in two monkeys (M1, M2) using two single electrodes (92 sessions; 235 IT units, 69 LFP sites; 170 FEF units, and 92 LFP sites; see Methods section). All of the FEF and IT units in the analysis had overlapping RFs, both responding to visual stimuli in the FEF RF. To examine the neural basis of object WM, we trained two monkeys to perform an object delayed-match-to-sample (DMS) task (Fig. 1a) in which they had to remember the identity of a sample stimulus throughout a delay period (1 s). The sample stimulus appeared either within the FEF RF (In condition) or 180 degrees away (Out condition). Following the delay period, two stimuli appeared in locations rotated 90° relative to the sample location, and monkeys were rewarded for making a saccadic eye movement to the one that matched the sample stimulus, regardless of its location. Different sample objects evoked a comparatively stronger or weaker response in the IT unit being recorded, which was used to define the preferred (Pref) and non-preferred (NPref) stimulus conditions. Performance of the two animals is shown in Fig. 1b (correct trials = 70 ± 1% M1, 77 ± 1%

M2; 51 sessions in M1 and 41 sessions in M2). The Supplementary Information contains information on the relationship between performance and reaction times (Fig. S1a), saccade landing points (Fig. S1b), and sample eccentricity (Fig. S1c), along with RF positions (Fig. S1d), eye position for In vs. Out and correct vs. wrong trials (Fig. S1e), and microsaccade rate for correct vs. wrong trials (Fig. S1f); neither eye position nor microsaccade rate was related to performance. The results for the two monkeys separately are provided in Table S1. As in previous studies[14-16], we focused on visual stimuli that evoked the strongest visual responses in IT, as they are more likely to be involved in maintaining the memory of those stimuli. There was a correlation between object selectivity during the visual and delay periods (Fig. S3). Therefore, we analyzed the Pref condition for IT, and the sample In condition for the FEF. For analyses involving both FEF and IT, we considered the Pref In condition, unless otherwise specified.

**Within-area neural activity does not predict behavioral performance**. We first examined whether changes in the FEF or IT firing rate were correlated with behavioral performance on the object WM task. Consistent with an earlier study[17], the population of FEF units (102 M1, 68 M2) exhibited persistent activity, measured as an increase in normalized firing rate (NFR) during the delay period compared to their baseline ($\Delta$NFR = 0.066 ± 0.011, $p < 10^{-10}$, $n = 170$; Fig. 1c, upper left). This persistent activity was significant in 58% of FEF units (99/170). This increase in delay-period FEF spiking activity was only slightly greater during correct (Cr) trials than wrong (Wr) trials ($\Delta$NFR = 0.008 ± 0.003, $p = 0.013$, $n = 170$; Figs. 1c, upper right and S4a). In the population of IT units (171 M1, 64 M2), although 30% of IT units (70/235) exhibited significantly elevated delay period spiking activity vs. baseline, in the whole population the increase in delay period firing rate was not significant ($\Delta$NFR = −0.005 ± 0.007, $p = 0.465$, $n = 235$). Moreover, there was no significant difference in the delay period spiking between the correct and wrong trials ($\Delta$NFR = −0.002 ± 0.005, $p = 0.982$, $n = 232$; Figs. 1c, lower right and S4b). Delay period activity in FEF and IT was selective for sample location (In-Out) and object identity (Pref-NPref), respectively (FEF spatial selectivity = 0.059 ± 0.011, $p < 10^{-8}$, $n = 170$; IT object selectivity = 0.046 ± 0.005, $p < 10^{-15}$, $n = 235$). However, the magnitude of this selectivity did not vary with performance ($\Delta$FEF spatial selectivity = 0.000 ± 0.007, $p = 0.562$, $n = 170$, Fig. S4c; $\Delta$IT object selectivity = −0.016 ± 0.009, $p = 0.080$, $n = 232$, Fig. S4d). Thus, neither FEF's nor IT's average spiking activity was a strong predictor of object WM performance.

Next, we investigated the strength of neural oscillations within each area. We calculated a time-frequency map of the LFP power spectra for each of the FEF sites (51 M1, 41 M2) and IT sites (35 M1, 34 M2 after exclusions due to noise or lack of object selectivity) using a wavelet transform (see Methods section). All effects were tested in the theta ($\theta$, 4–8 Hz), alpha ($\alpha$, 8–15 Hz), beta ($\beta$, 22–28 Hz), low-gamma (L$\gamma$, 35–50 Hz), and high-gamma (H$\gamma$, 51–130 Hz) frequency bands. During the delay period, FEF power in the high-gamma band was greater than baseline, while theta, alpha, beta, and low-gamma power was lower (see Table S2 and Supplementary Information). However, there was no significant difference in LFP power between correct and wrong trials during the delay period for any frequency band ($\Delta$power$_\theta$ = −0.037 ± 0.030, $p = 0.160$, $\Delta$power$_\alpha$ = 0.007 ± 0.026, $p = 0.277$, $\Delta$power$_\beta$ = −0.024 ± 0.027, $p = 0.173$, $\Delta$power$_{L\gamma}$ = −0.029 ± 0.031, $p = 0.061$, $\Delta$power$_{H\gamma}$ = −0.018 ± 0.030, $p = 0.104$, $n = 92$ sites; Figs. 1d, top and S4e). Within IT, there was a decrease in power in the high and low-gamma bands during the delay period compared to baseline. Only high-gamma band power reflected sample identity

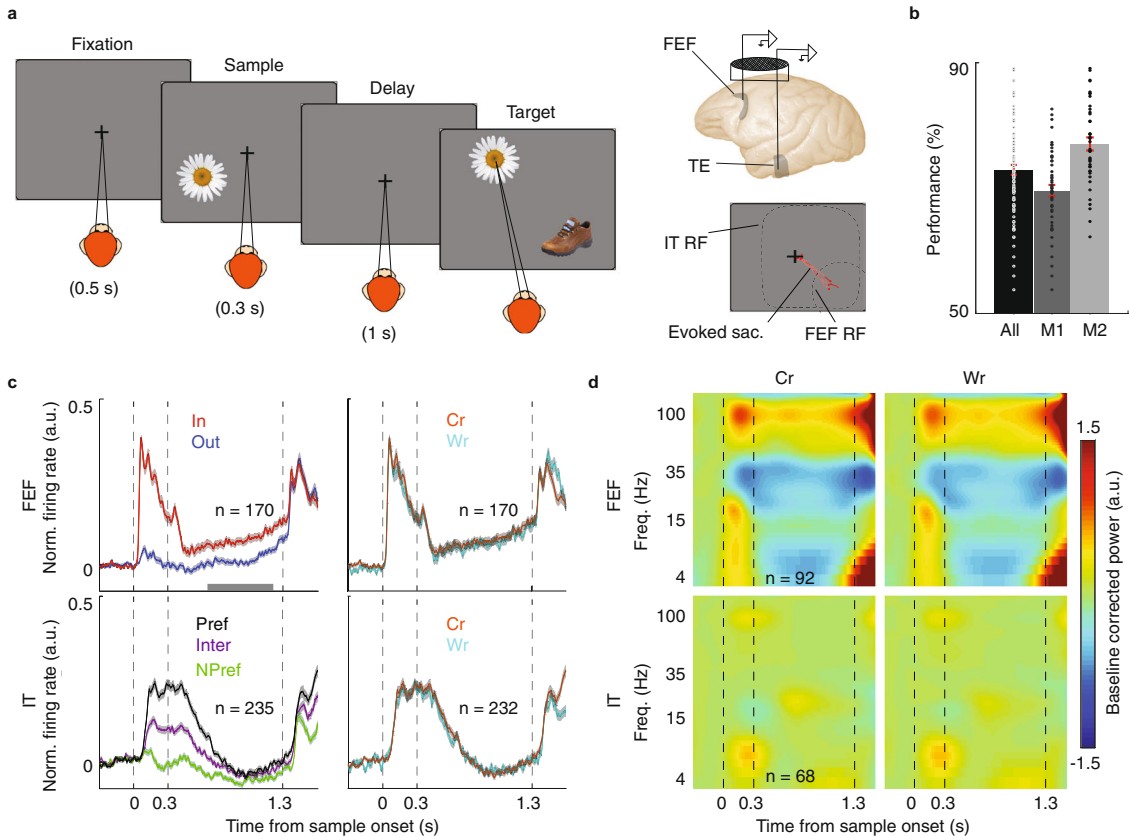

**Fig. 1 Firing rate and LFP power in IT and FEF were largely unrelated to DMS performance. a** Schematic of the object DMS task. Left, After the monkey fixated, a sample stimulus appeared either in or opposite the FEF neuron's RF location. The sample stimulus disappeared, and the monkey maintained fixation throughout a blank delay period. The same stimulus appeared again (target) along with a distracter and the monkey needed to saccade to the target stimulus to receive a reward. Right, Recordings were made simultaneously from the FEF and the TE part of IT in a single recording chamber. The FEF RF was estimated based on microstimulation-evoked saccades (red lines). The IT RF generally encompassed both sample locations. **b** Performance of individual monkeys (M1, M2) and their average (All) on the DMS task; points show mean performance in individual sessions; bars show mean across sessions ± SEM in red (n = 51 sessions in M1 and 41 sessions in M2). **c** Normalized firing rates of FEF (top) and IT populations (bottom) across the course of the DMS task, when the sample was inside (red) or outside (blue) of the neurons' RFs, and when the sample object was preferred (black), non-preferred (green), or intermediate (violet) are shown on the left. Right, normalized firing rate of the FEF (top) and IT populations (bottom) for correct vs. wrong trials. Data (mean ± SEM) were smoothed within a window of 10 ms. **d** Time-frequency maps of normalized LFP spectral power for FEF (top, n = 92 LFP sites) and IT (bottom, n = 68 LFP sites) for correct (left) and wrong (right) trials.

(see Table S2 and Supplementary Information). However, as in the FEF, there was no significant difference in high-gamma band LFP power between correct and wrong trials during the delay ($\Delta$power $= 0.002 \pm 0.008$, $p = 0.112$, $n = 68$ sites; Fig. 1d, bottom). Nor did power in any other frequency band reflect performance (Table S2 and Fig. S4f). Overall, although FEF and IT firing rates and LFP power during the delay period were modulated by the sample object and location, none of these measures were strongly predictive of behavioral performance.

**Inter-areal beta band coupling predicts performance and reflects the content of WM.** Since neither spiking nor LFP activity within FEF or IT was strongly predictive of performance, we next investigated the synchronization between FEF and IT sites in our search for behavioral correlates of WM. As a measure of inter-areal coupling, we calculated PPL between simultaneously recorded LFPs from FEF and IT (35 M1, 34 M2); PPL was calculated across time and frequency, and shuffle corrected (see Methods section). Importantly, for correct trials we observed a significant enhancement in beta band PPL during the delay period compared to baseline ($\Delta$PPL $= 0.267 \pm 0.108$, $p = 0.035$, $n = 69$; 37 out of 69 pairs showed a significant increase, Fig. 2a),

indicating a more consistent phase relationship between beta LFPs in FEF and IT during successful memory maintenance, and a significant decrease in wrong trials ($\Delta$PPL $= -0.211 \pm 0.095$, $p = 0.028$, $n = 63$; 39 out of 69 pairs showed a significant decrease). An increase in alpha and beta phase locking between areas also occurred during the sample period compared to baseline (see Supplementary Information). There was no delay-period increase in PPL for other frequency bands (Table S2). We compared baseline-adjusted delay period PPL for correct vs. wrong trials, and found that in the beta band, PPL was greater for correct trials ($\Delta$PPL $= 0.471 \pm 0.134$, $p < 0.001$, $n = 63$; Figs. 2a–c and S5b). This enhanced beta band PPL for correct trials was significant after trial matching or without the shuffle correction (Fig. S5a). Successful WM performance was thus more strongly associated with the phase locking between the FEF and IT than with the average spiking activity or the power of LFPs within either area.

Having identified the beta coupling between areas as a predictor of successful WM maintenance, we next examined whether this measure also related to the content of WM. To do this, we compared the beta band PPL between areas for different sample objects and locations. Measuring the object selectivity (Pref, In – NPref, In), we found that the beta band PPL between the FEF and IT carried

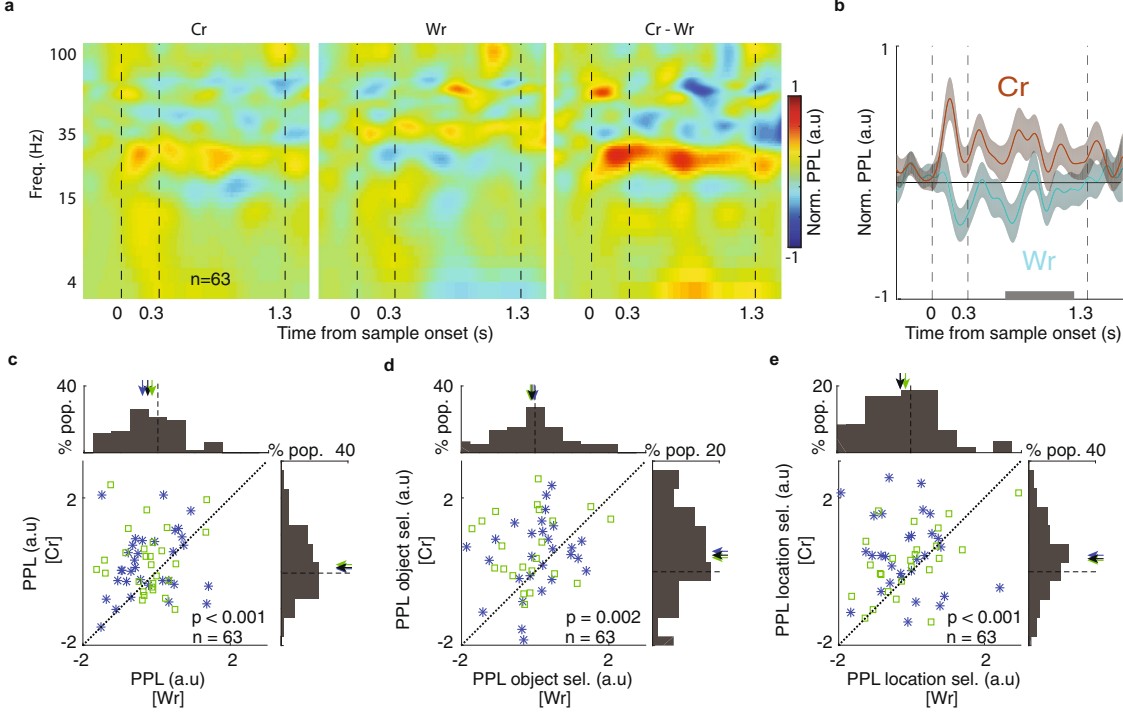

**Fig. 2 Inter-areal beta coupling during correct and wrong trials. a** PPL between FEF and IT was different for Cr and Wr trials. The time-frequency map of PPL, normalized to baseline across the course of the DMS task for correct (left), wrong (middle), and the difference between Cr and Wr trials (right). **b** The beta band PPL during sample and delay differed between Cr and Wr trials. Plot shows the average of the beta band PPL (mean ± SEM across recorded pairs) across the course of the DMS task for the correct (brown) and wrong (cyan) trials. Gray bar indicates the analysis window for **c–e**. **c** Comparison of the beta band PPL during the delay period for correct versus wrong trials. The beta band PPL was significantly higher for correct trials ($n = 63$ pairs, $p = 0.0009$, Wilcoxon's signed-rank test, two-sided). M1 plotted in blue and M2 in green. **d** Comparison of object selectivity of the beta band PPL during the delay period for correct versus wrong trials. Object selectivity of the beta band PPL was significantly higher for correct trials ($n = 63$ pairs, $p = 0.0022$, Wilcoxon's signed-rank test, two-sided). **e** Comparison of spatial selectivity of the beta band PPL during the delay period of correct versus wrong trials. Spatial selectivity of the beta band PPL was significantly higher for correct trials ($n = 63$ pairs, $p = 0.0004$, Wilcoxon's signed-rank test, two-sided).

information about the identity of remembered objects (object selectivity = $0.483 \pm 0.163$, $p = 0.002$, $n = 69$). No other frequency band showed similar significant object selectivity of the PPL (Table S2). Moreover, the object selectivity of the beta band PPL was greater during correct compared to wrong trials ($\Delta$object selectivity = $0.654 \pm 0.173$, $p = 0.002$, $n = 63$; Figs. 2d and S5c). Thus, the content of object WM was reflected in the beta band PPL between the FEF and IT, and the strength of this encoding was correlated with behavioral performance. We also investigated whether beta coupling encodes the sample location during the delay period by measuring its spatial selectivity (Pref, In − Pref, Out). We found that the beta band PPL during the delay period also carried spatial information (spatial selectivity = $0.394 \pm 0.142$, $p = 0.003$, $n = 69$, Fig. S5d). In addition, the spatial selectivity of the beta band PPL was greater for correct compared to wrong trials ($\Delta$spatial selectivity = $0.633 \pm 0.166$, $p < 0.001$, $n = 63$; Fig. 2e). The difference in beta band PPL was correlated with the magnitude of spatial selectivity in FEF firing rate ($r = 0.220$, $p < 10^{-5}$, $n = 136$; Fig. S6 and Table S1). Thus, beta coupling between the FEF and IT reflected both the sample identity and location of the remembered stimulus, and both the spatial and object selectivity were stronger on correct trials.

In order to investigate the directionality of fronto-temporal coordination, we next examined the phase difference between FEF and IT LFPs during the delay period. On correct trials, we observed a positive phase difference between FEF and IT, indicating that FEF led IT; Fig. 3a shows a distribution of phase differences between FEF and IT for correct (top) and wrong (bottom) trials. The FEF beta phase led IT on correct trials ($\mu = 26.856°$, $\kappa = 0.997$, $p < 10^{-6}$, $n = 63$, Von Mises test), but not on

wrong trials ($\mu = -6.688°$, $\kappa = 0.511$, $p = 0.003$, $n = 63$, Von Mises test). The variability in phase difference values across sessions was also reduced on correct trials ($\kappa_{Cr} = 1.037 \pm 0.241$, $\kappa_{Wr} = 0.553 \pm 0.183$, $p = 0.008$, $n = 1001$x bootstrap); this difference was only present during the delay period (Fig. 3b). These results suggest that the FEF plays a leading role in the beta band coupling during correct trials.

**Beta-band phase modulates spiking activity in IT.** Since beta band phase-phase coupling between areas was associated with successful memory maintenance, we next tested whether this beta phase modulated spiking activity. We analyzed whether the increase in beta band PPL was accompanied by an increase in inter-areal spike-phase locking (SPL). To calculate SPL, each spike was assigned a phase value based on the simultaneously recorded LFP; SPL was the circular average of all spike phases at a particular frequency (see Methods section). We calculated SPL for both FEF spikes and IT LFPs, and IT spikes with FEF LFPs, but only the latter showed significant correlations with behavior (Fig. 4). Figure 4a illustrates SPL between FEF LFPs and IT spikes for correct vs. wrong trials across various frequencies; there was an increase in the theta, beta, and low-gamma band SPL on correct compared to wrong trials ($\Delta$SPL$_\theta = 0.005 \pm 0.004$, $p = 0.034$; $\Delta$SPL$_\beta = 0.007 \pm 0.003$, $p < 0.001$; $\Delta$SPL$_{L\gamma} = 0.009 \pm 0.003$, $p = <0.001$; $n = 221$). In addition, there was an increase in the beta band SPL between FEF LFPs and IT spikes during the Pref, In compared to the NPref, In condition, indicative of the object selectivity of this SPL ($\Delta$SPL = $0.007 \pm 0.003$, $p = 0.003$, $n = 235$). In contrast, neither the beta SPL between FEF spikes and IT LFPs, nor within-IT beta SPL, nor within-FEF beta SPL was predictive of

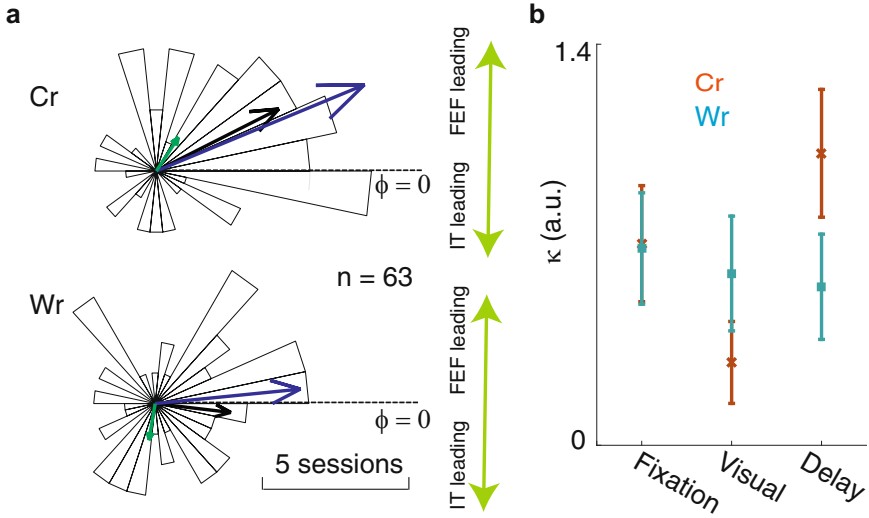

**Fig. 3 Inter-areal beta coupling led by FEF. a** Beta-band phase differences between areas reflected performance. The polar plot shows a histogram of phase differences between the FEF and IT LFPs in the beta band for correct trials (top) and wrong trials (bottom, $n = 63$ pairs). Arrows show the circular means across all pairs, for M1 (blue), M2 (green), and both (black). **b** Reliability of phase difference values ($\kappa$) across sessions (mean ± SD over bootstrap distribution; bootstrap $n = 1001$) during the fixation, visual, and delay periods for Cr (brown) and Wr (cyan) trials.

performance (Fig. 4b–d). There was no difference in the beta band SPL between FEF spikes and IT LFPs for correct vs. wrong trials ($\Delta SPL = 0.006 \pm 0.004$, $p = 0.167$; $n = 120$; Fig. 4d). The beta band SPL between IT LFPs and IT spikes was not significantly different for correct vs. wrong trials ($\Delta SPL = 0.006 \pm 0.003$, $p = 0.088$, $n = 182$ neuron- LFP pairs; Fig. 4b), although there was an increase in the theta and low-gamma band SPL on correct compared to wrong trials ($\Delta SPL_\theta = 0.007 \pm 0.004$, $p = 0.035$; $\Delta SPL_{L\gamma} = 0.006 \pm 0.003$, $p = 0.003$; $n = 182$). For FEF spikes and LFPs, only the high-gamma band SPL was significantly different for correct vs. wrong trials ($\Delta SPL = 0.006 \pm 0.002$, $p = 0.002$, $n = 170$ neuron- LFP pairs; Fig. 4c). Thus, inter-areal, but not within area, beta coupling (PPL and SPL) predicted memory performance. The inter-area specificity of these signatures, and the directionality of the SPL effect (FEF LFP to IT spikes only), suggests a model in which FEF beta activity modulates IT spiking.

**Beta coupling between the FEF and IT as a predictor of IT's object discriminability.** Object-selective persistent activity in IT has often been suggested as a basis of WM maintenance[15]. Beta coupling can also reflect the identity of the object held in WM[9,18]. We sought to determine if there was a relationship between beta coupling and the object-selective persistent activity in IT. To test for this relationship, we quantified beta coupling on individual trials (see Methods). We sorted trials by beta band PPL value and designated the third of trials with the highest PPL as "High" and the lowest third as "Low" trials (Methods). Using ROC analysis, we calculated the time-course of the object discriminability of IT spiking activity for High and Low PPL trials (Fig. 5a). For the In condition, High PPL trials had greater object discriminability than Low PPL trials during the delay period ($\Delta AUC = 0.047 \pm 0.021$, $p = 0.028$ $n = 74$;). For the Out condition, there was lower object discriminability on High vs. Low PPL trials ($\Delta AUC = -0.044 \pm 0.021$, $p = 0.042$, $n = 73$, Fig. S7a). This effect is not merely a result of increased beta power within an area: if trials were categorized by beta band power of either IT or FEF LFPs, there is no difference between object discriminability for High vs. Low power trials (IT beta power, $\Delta AUC_{In} = -0.006 \pm 0.021$, $p = 0.489$, $n = 75$; FEF beta power, $\Delta AUC_{In} = 0.009 \pm 0.020$, $p = 0.823$, $n = 72$, Fig. 5b, c, IT beta power, $\Delta AUC_{Out} = 0.001 \pm 0.018$, $p = 0.954$, $n = 75$; FEF beta power, $\Delta AUC_{Out} = 0.029 \pm 0.019$, $p = 0.274$, $n = 72$, Fig. S6b, c).

Figure 5d shows the modulation index (MI) of object discriminability values (see Methods) for High vs. Low PPL trials for object-selective IT units, across time for the In (red) and Out (blue) conditions. There was an increase in the modulation of object discriminability by PPL during the delay period for the In condition compared to the Out condition ($\Delta MI = 0.075 \pm 0.021$, $p < 0.001$, $n = 73$; Fig. 5e). The modulation index did not differ based on beta power within IT ($\Delta MI = -0.006 \pm 0.023$, $p = 0.951$, $n = 74$) or FEF ($\Delta MI = -0.022 \pm 0.022$, $p = 0.510$, $n = 71$). We next sought to test whether this modulation by beta coupling was dependent on the strength of object discriminability of the IT unit. Across the IT population, for the In condition, there was a correlation between IT units' object discriminability and the discriminability difference for High vs. Low PPL trials ($r = 0.100$, $p = 0.033$, $n = 159$; Fig. 5f). No such correlation was observed for the Out condition ($r = 0.013$, $p = 0.421$, $n = 159$; Fig. S7d), or based on beta power within IT ($r = 0.051$, $p = 0.161$, $n = 160$) or FEF ($r = 0.048$, $p = 0.186$, $n = 160$). These results support the idea that beta coupling between the FEF and IT, unlike power within either area alone, predicts the maintenance of object identity within IT cortex.

## Discussion

Prefrontal areas are known to play an important role in WM. In order to understand how PFC interacts with sensory areas during WM, we simultaneously recorded from a prefrontal area (FEF) where neurons exhibit spatially selective persistent activity and from IT, where neurons exhibit object-selective persistent activity, during an object WM task (Fig. 1). Whereas neither the average spiking activity nor the LFPs within either of these areas strongly predicted behavioral performance (Fig. 1c, d), the phase-phase locking between the two areas, specifically in the beta frequency range, was robustly correlated with the behavioral outcome (Fig. 2a–c). This coordination between areas was selective for both the spatial and object content of WM (Fig. 2d, e). FEF beta oscillations led those in IT (Fig. 3a) and modulated the timing of its spiking activity (Fig. 4), and both of these effects were stronger on correct trials. This inter-areal oscillatory coherence not only predicted the behavioral outcome, it also predicted the strength of object discriminability within IT during memory maintenance (Fig. 5). Because oscillatory coherence predicted object selectivity in IT (Fig. 5), it is likely that with sufficient data IT object-selective

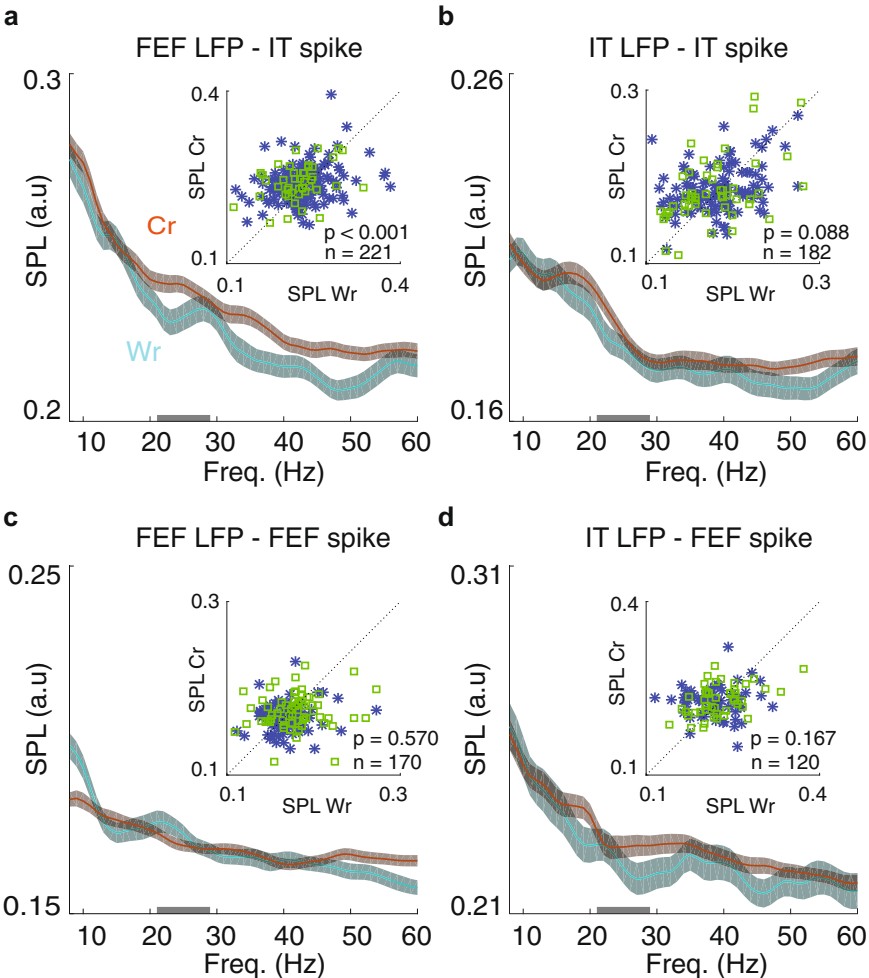

**Fig. 4 IT spiking activity, coupled to FEF phase, reflected performance. a** The beta band SPL between IT spikes and FEF LFPs reflected performance. Plots show the SPL for correct (brown) and wrong (cyan) trials across frequencies for all pairs of IT units and simultaneously recorded FEF LFPs. Gray bar on x-axis indicates the beta frequency range. Data were smoothed within a window of 1 Hz and represented as mean ± SEM. Inset scatter plots compare the beta band SPL for correct vs. wrong trials ($n = 221$ pairs, $p = 0.0009$, Wilcoxon's signed-rank test, two-sided). M1 plotted in blue and M2 in green. **b–d**, Plots show the SPL for correct and wrong trials across frequencies for all pairs of IT units and IT LFPs (**b**, $n = 182$ pairs, $p = 0.088$, Wilcoxon's signed-rank test, two-sided), FEF units and FEF LFPs (**c**, $n = 170$ pairs, $p = 0.570$, Wilcoxon's signed-rank test, two-sided), and for FEF units and IT LFPs (**d**, $n = 120$ pairs, $p = 0.167$, Wilcoxon's signed-rank test, two-sided). Smoothing and scatter plot as in **a**.

activity would also be correlated with performance (as FEF activity was); however, these data show that the inter-areal signatures are a more robust indicator of neural mechanisms related to WM performance.

It is important to note that multiple sources of errors are possible during the object DMS task. Encoding errors could result from lapses in attention during the sample period, or if attention was incorrectly directed to the alternative sample location. During the delay period, there could be failures of maintenance[19], or interference from samples remembered during previous trials[20]. Effects of recent reward history, on the other hand, presumably influence choice during the target selection stage[21]. Errors in target selection could arise due to impulsivity or selection biases, either based on recent reward history or more persistent biases in target location or object identity. Despite this heterogenous mix of sources of errors, we were nevertheless able to identify inter-areal signatures that correlated with memory performance during the delay period. Observed differences between correct and wrong trials during the delay period may either reflect failures in memory, or failures to encode the stimulus. Clearly, if a sample is not encoded, it cannot be remembered, and thus error trials caused by failures of encoding should exhibit no memory-related

activity. Errors in target selection following successful encoding and memory maintenance should result in mislabeling of trials with sufficient memory activity as 'wrong' trials, weakening the strength of behavioral correlations. Nevertheless, we observe significant behavioral correlations for neural signatures during the delay period, suggesting that either the magnitude of the effects we observed would be stronger when limiting analysis only to trials with failures of memory, or that there were comparatively few target selection errors.

Our results support the idea that the interaction between prefrontal and sensory areas contributes to WM maintenance. Although we cannot rule out the possibility that one or more other areas drive WM selective signals in both the FEF and IT during WM, the findings that FEF leads IT in phase, and that inter-areal coherence predicts IT object selectivity, are consistent with a direct gating of object-selective persistent activity in IT by the FEF. This leading role of the FEF dovetails with previous results showing Granger-based causality of the FEF on IT during memory maintenance, but not the reverse[22]. The known influence of FEF neurons on the visual responses in visual cortex[23–25], and the evidence that FEF neurons send memory-related spiking activity directly to visual areas[26], further support the idea that

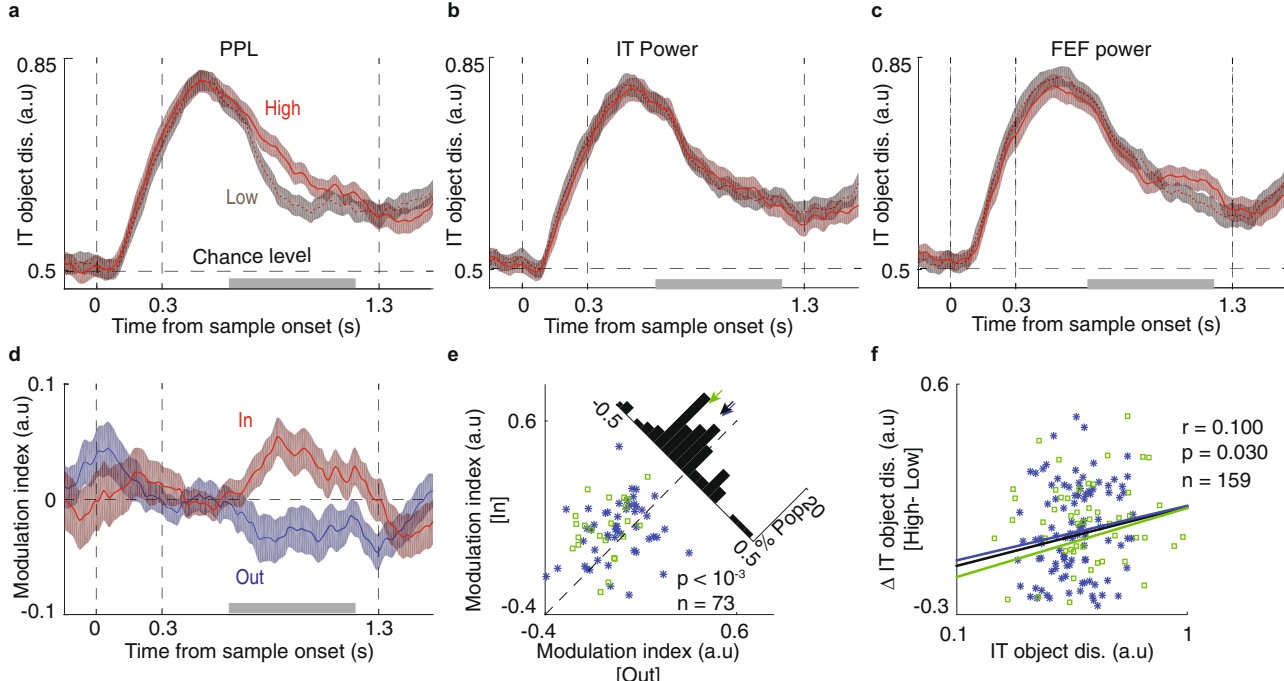

**Fig. 5 Enhancement of object coding by IT spiking activity during high beta PPL. a** Object discriminability was greater on trials with high beta PPL. Time-course of the object discriminability values for High (light red) vs. Low (dark red) PPL trials for object-selective IT units ($n = 74$ units). Data were smoothed within a window of 1 ms and represented as mean ± SEM. Gray bar in **a–d** indicates portion of delay period used for analysis in **e**, **f**. **b**, **c** Object discriminability does not reflect beta power within IT (**b**, $n = 75$ units) or FEF (**c**, $n = 72$ units). Time-course of the object discriminability values for High (light red) vs. Low (dark red) beta band power trials for object-selective IT units. Smoothing and error bar as in **a**. **d** Plot shows the time-course of the modulation index of object discriminability (High vs. Low PPL trials), for In (red) and Out (blue) conditions ($n = 73$ units). Data were smoothed with a window of 1 ms and represented as mean ± SEM. **e** Scatter plot illustrates the modulation index of object discriminability (High vs. Low PPL) during the delay period for In vs. Out conditions. The histogram in the upper right shows the difference between In and Out ($n = 73$ units, $p = 0.00018$, Wilcoxon's signed-rank test, two-sided). M1 plotted in blue and M2 in green. **f** Scatter plot shows the correlation between discriminability for each IT unit (x-axis) and the difference in discriminability for that unit between high and low beta PPL trials (y-axis, $n = 159$ units, Kendall correlation, $r = 0.100$, $p = 0.030$). Lines show fits for individual monkeys (blue, green) and combined data (black).

FEF spiking activity has a direct impact on IT during WM. It is important to note that this is not the first time a spatial signal has been shown to enhance feature selectivity. In the case of covert spatial attention[27,28], spatial cues have been shown to increase feature information in V4[29,30] and IT[31]. A similar gating of feature information by a spatial signal is observed when attention is exerted overtly, when the representations of targets of upcoming eye movements are enhanced[32–35]. More specifically, FEF spiking activity, lacking robust feature selectivity, has been shown to contribute to feature selectivity in these areas[23–25,36,37]. A spatial tag has also been suggested as a basis for binding object features[38,39], including during short term memory[40,41]; consistent with this hypothesis, retroactive spatial cues can improve object memory[42–46]. In our data, coherence between IT and the FEF could reflect the deployment of spatial attention for feature binding during object memory maintenance[40,41,47,48]. However, in the absence of a competing stimulus, this FEF contribution might not be apparent[49]. These results suggest that perhaps, similar to the case of spatial attention (or indeed due to a role of spatial attention in memory maintenance[44,50,51]), a non-selective signal highlighting relevant information is sufficient to boost object-selective spiking activity in IT.

The coordination between the FEF and IT occurred mostly within the beta band, and the consistency of modulations in only this band across multiple measures (e.g. SPL, PPL) demonstrates the robust involvement of this frequency band. Moreover, given their level of significance, the beta band effects remain significant even when subjected to multiple comparison tests (e.g.

Bonferroni's correction, see Table S1). In prefrontal and visual areas, increased beta power has been reported during the delay period of a variety of tasks[9,52–58], leading to the suggestion that maintenance of the current state, whether motor or cognitive, is the unifying principle of beta oscillations[59]. Indeed, beta band coherence between IT sites is correlated with object WM performance in humans[57]. Such a content-agnostic role for beta oscillations in maintaining current states and information, whatever those might be, is consistent with the idea of a non-selective gain signal from the FEF gating object information within sensory areas. Note that despite the lack of feature selectivity in the FEF, we nevertheless observed spatial and feature selective coordination between the FEF and IT (Fig. 2d-e and Fig. S4). The finding that an area without feature selectivity can develop content-specific coordination with a feature selective sensory area is further evidence that non-selective signals can nevertheless gate both selective sensory and memory spiking activity.

## Methods

**Animals and recordings**. Two adult male rhesus monkeys (10 and 11 kg) were used in this study. M1's experiments were performed in the School of Cognitive Science at IPM, and M2's experiments were performed at Stanford University. All experimental procedures were in accordance with the National Institutes of Health Guide for the Care and Use of Laboratory Animals and the Society for Neuroscience Guidelines and Policies. The protocols for all experimental, surgical, and behavioral procedures were approved by the Institute Fundamental Science committee for M1, and by the Stanford University Institutional Animal Care and Use Committee for M2. Structural magnetic resonance imaging (M1, M2) and CT scan

(M1) were performed to locate the arcuate sulcus and prelunate gyrus for the placement of a recording chamber in a subsequent surgery. All surgical procedures were carried out under Isoflurane anesthesia and strict aseptic conditions. Prior to undergoing behavioral training, each animal was implanted with a stainless-steel custom-made chamber, attached to the skull using orthopedic titanium screws and dental acrylic. For M1, within the 30 × 70 mm chamber a craniotomy was performed (5 mm to 30 mm A/P, and 0 mm to 23 mm M/L). For M2, a craniotomy was performed within the a 20 mm diameter circular chamber (centered at 23 mm M/L and extended from 25 mm A/P to 5 mm A/P).

**Behavioral monitoring.** Animals were seated in custom-made primate chairs, with their head restrained and a tube to deliver juice rewards placed in their mouth. For M1, eye position was monitored with an infrared optical eye tracking system (EyeLink 1000 Plus Eye Tracker, SR Research Ltd, Ottawa CA), with a resolution of <0.01° RMS; eye position was monitored and stored at 2 KHz. The EyeLink PM-910 Illuminator Module and EyeLink 1000 Plus Camera (SR Research Ltd, Ottawa CA) were mounted in front of the monkey, and captured eye movements. Stimulus presentation and juice delivery were controlled using custom software, written in MATLAB using the MonkeyLogic toolbox[60]. Visual stimuli were presented on an LED-lit monitor (Asus VG248QE: 24in, resolution 1920 × 1080, 144 Hz refresh rate), positioned 65.5 cm in front of the animal's eyes. A photodiode (OSRAM Opto Semiconductors, Sunnyvale CA) was used to record the actual time of stimulus appearance on the monitor. For M2, eye position was monitored with a scleral search coil and digitized at 500 Hz (CNC Engineering, Seattle, WA). The spatial resolution of eye position measurements was «0.1 deg. Stimulus presentation, data acquisition and behavioral monitoring were controlled by the CORTEX system. Visual stimuli were presented on a 29 × 39 colorimetrically calibrated CRT monitor (Mitsubishi Diamond Pro 2070SB-BK) with medium short persistence phosphors. A photodiode (OSRAM Opto Semiconductors, Sunnyvale CA) was used to record the actual time of stimulus appearance on the monitor, with a continuous signal sampled and stored at 32 KHz.

**Behavioral tasks**

*Eye calibration task.* The fixation point, a ~0.25 dva white circle, appeared either centrally or displaced by ±10dva in the horizontal or vertical axis, and the monkey maintained fixation within a ± 1.5 dva window for 1.5 s to receive a reward.

*A rapid serial visual presentation (RSVP) task.* The RSVP task was used to quickly screen 44 sample objects and identify at least one which evoked IT visual responses for subsequent use in the DMS task[61,62]. Monkeys fixated within a ± 1.5 dva window around the central fixation point, and 10 objects (from 44 possible) were pseudo-randomly presented for 150 ms each with 100 ms between stimuli. Responses from the recording site in IT cortex were monitored audibly and visually by the experimenter. Stimuli evoking the highest response, an intermediate response, and little or no response were selected for use in the subsequent DMS task; final assignment of objects as Pref/Inter/NPref for analysis was based on responses during the DMS task. Only three objects were used per recording session in order to maximize the number of trials (and statistical power) for each condition.

*RF estimation tasks.* We identified FEF sites by eliciting short-latency, fixed-vector saccadic eye movements with trains (50–100 ms) of biphasic current pulses (50 mA; 250 Hz; 0.25 ms duration). RF estimation was conducted by having the monkey fixate within a ±1.5 dva window around the fixation point (central or displaced by ±10 dva), while stimulation was applied to evoke a saccade. The average endpoint of evoked saccades (landing points) was used to estimate the center of the FEF RF.

By design, all of the FEF and IT units recorded had overlapping RFs. IT RFs are large[63], and typically include the low eccentricity (<13 dva) contralateral visual hemifield in which all our FEF RFs were centered. Both FEF and IT neurons responded to stimuli placed in the estimated FEF RF (Fig. 1c).

*Object DMS task.* The DMS task is a widely used WM task; the version we use requires object WM. Monkeys fixated within a ± 1.5 dva window around the central fixation point. After 500 ms of fixation, a 3 dva sample object was presented either inside the FEF RF, or 180° away in the opposite hemifield, and remained onscreen for 300 ms. The animal then remembered the sample identity while maintaining fixation for 1 s (memory period) before the central fixation point was removed and two targets appeared (the sample object and a distractor). Target positions were rotated 90 degrees relative to the sample location, so no target appeared at the sample location, nor did the sample location predict the correct target location. The monkey had to shift its gaze to a ± 4 dva window around the sample object to receive a juice reward. If the monkey completed the trial but selected the distractor object, it was labeled a wrong trial. Note that in this version of the task, the position of the sample is irrelevant for task performance. In designing the task, we intentionally chose a long sample duration (300 ms) in hopes of minimizing encoding errors (estimated 50 ms/sample needed for encoding each item[64]).

**Neurophysiological recording.** Two electrodes were mounted on a single recording chamber and positioned within the craniotomy area using two Narishige two-axis platforms allowing continuous adjustment of the electrodes' position. Two 28-gauge guide tubes were lowered to contact or just penetrate the dura, using a manual oil hydraulic micromanipulator (Narishige, Tokyo, Japan). For M1 and FEF of M2, varnish-coated tungsten microelectrodes (FHC, Bowdoinham), shank diameter 200–250 μm, impedance 0.2–1 MΩ (measured at 1 kHz), were advanced into the brain for the extracellular recording of neuronal activity. For IT recordings in M2, a 32 gauge (235 mm outer diameter) guide tube was advanced ~15 mm through the brain (at a rate of ~0.75 mm/minute) by a custom-modified electronic motor-driven microdrive, stopping at or just above the upper bank of the Superior Temporal Sulcus. An electrode (75–100 μm diameter) was then advanced another 5–12 mm using a hydraulic microdrive (Narashige); high compressive strength was necessary for the electrode to exit the guide tube without bending. Activity was recorded extracellularly with varnish-coated tungsten microelectrodes (FHC) of 0.2–1.0 MΩ impedance (measured at 1 kHz). For M1, single-electrode recordings used a pre-amplifier and amplifier (Resana, Tehran, Iran), filtering from 300 Hz–5 KHz for spikes and 0.1 Hz–9 KHz for LFPs. Spike waveforms and continuous data were digitized and stored at 40 kHz for offline spike sorting and data analysis. Extracellular waveforms were digitized and classified as single neurons using manual offline sorting (FHC, Plexon). We also used cross-validation classifier performance to define a criterion for the accuracy of the well-isolated clusters[65]. This was done by training a binary support vector machine (SVM) classifier using 70% of sorted spikes with an equal number of samples for each pair of clusters from a given recording site. The average classifier performance for the remaining 30% of samples across cluster pairs was taken as the isolation quality of sorted neurons in each session. The clusters of spikes with high isolation quality (>90%) were selected as well-isolated units[66] (Fig. S2). Area IT was identified based on stereotaxic location, position relative to nearby sulci, patterns of gray and white matter, and response properties of units encountered. Area FEF was identified based on stereotaxic location, position relative to acute sulcus, patterns of gray and white matter, and also microstimulation. By single-electrode exploration and electrical stimulation, areas posterior of the arcuate sulcus (evoking movements of hand and face) and the FEF (evoking saccadic eye movements) were identified within the recording chamber prior to beginning data collection.

Data are reported from 92 simultaneous FEF and IT recordings (51 M1, 41 M2). For IT spiking data, Pref/Inter/NPref objects were determined based on the peak response across the visual and delay periods[67] during the DMS task. For IT LFP data, object selectivity in multiunit responses at each recording site was used to determine the suitability of the LFP data (similar to[67], if visual responses for the Inter condition were not significantly different from Pref or NPref activity, the delay period response was used to define the Pref/NPref object). As a result, LFP data from 8 IT sites (2 M1, 6 M2) were excluded due to lack of object-selective multiunit activity, and LFP data from 15 IT sites (14 M1, 1 M2) were discarded due to insufficient SNR. When comparing correct and wrong trials, some LFP data and units were excluded due to low numbers of wrong trials in a given condition (<3 wrong trials); number of sites and units included in each analysis are reported in the text with the statistics.

**Data analysis.** All analyses were carried out with custom code written in Matlab (MathWorks). The raw LFP data was low-pass filtered (1–300 Hz) and resampled at 1 kHz. The LFP for each site was normalized by subtracting the mean and dividing by the s.d. (across all time points and trials). Analysis focused on the 600 ms window in the late delay (300–900 ms after stimulus offset), in order to exclude both the stimulus evoked response and presaccadic activity (~100 ms before target onset); this is the time window used for analysis unless otherwise stated. In the trial-matching procedure, the number of correct and wrong trials was matched by subsampling the larger distribution 1001 times, and taking the mean of that distribution.

For behavioral analysis we calculated performance based on the percentage of completed trials in which the monkey selected the correct target (aborted trials were not included in the analysis). Reaction times were calculated based on the time between target appearance and the eyes leaving the fixation window, and the Fig. S1a shows the median across trials for each session. Figure S1b shows the average of the standard deviations of saccade landing points for the two target positions for each session. Saccades and microsaccades were detected using a velocity threshold (20 dva/sec); microsaccades were defined as >0.1 dva and <1 dva.

For firing rate plots (Fig. 1c), individual spike trains were smoothed with a 10 ms gaussian kernel, and the mean response was calculated across trials for each condition. The normalized firing rate was calculated according to the formula NFR (t) = [Rate(t) − Baseline]/(MaxRate − Baseline), where Baseline was the average firing rate during the 500 ms of the fixation period before sample onset, and MaxRate was the maximum during the task.

To calculate LFP power and phase (Fig. 1d), we extracted the instantaneous amplitude and phase as a function of time and frequency by convolving the raw real-valued time series $x(t)$ with the complex Morlet wavelet $w(t, f_0)$ to obtain the complex output signal $y(t, f_0)$, also denoted as the analytic signal, where $f_0$ denotes the desired center frequency of the wavelet function. We used a value of $c = 7$ wavelet oscillations. The center frequencies $f_0$ were 1–130 Hz.

Time-frequency composition of the power spectrum was calculated by the amplitude of the analytic signal for each area. We averaged power values for each

condition. In order to remove the $1/f$ effect we calculated Z-transformed values for each trial by subtracting the mean of the baseline and dividing by the standard deviation of the baseline across sites for each condition. The time-frequency map was smoothed with a window of 2 Hz in frequency and 100 ms in time. Then, Z-transformed values were averaged across sites.

The power-power correlation was calculated over time and frequency. For each point $(f.t)$ the Kendall correlation between FEF power and IT power across trials was calculated for each condition. In order to compare the conditions, we calculated Z-transformed correlation values by subtracting the value during the baseline and dividing by the standard deviation across pairs for each condition.

In order to quantify within area phase locking value (PLV; Table S2), we considered the phase value of each FEF and IT LFP using phase calculated by wavelet. We calculated PLV for each time and frequency according to Eq. (1)

$$PLV(f.t) = \frac{1}{N}\left| \sum_{k=1}^{N} e^{i\left(\varphi_{fef/it}^{k}(f.t)\right)} \right| \quad (1)$$

where $\varphi_{fef}^{k}/\varphi_{it}^{k}$ were the instantaneous phase values in trial $k$ in frequency $f$ and time bin $t$, and $N$ was the number of trials. The PLV was in [0 1], where 1 indicates high phase locking and 0 is no phase locking. In order to compare the conditions, we calculated Z-transformed PLV values by subtracting its value during baseline and dividing by the standard deviation across pairs for each condition.

In order to quantify PPL between areas (Figs. 2 and S1), we considered the phase difference between the FEF and IT using phase calculated by wavelet. We calculated PPL for each time and frequency according to Eq. (2)

$$PPL(f.t) = \frac{1}{N}\left| \sum_{k=1}^{N} e^{i\left(\varphi_{fef}^{k}(f.t)-\varphi_{it}^{k}(f.t)\right)} \right| \quad (2)$$

where $\varphi_{fef}^{k}$ and $\varphi_{it}^{k}$ were the instantaneous phase values in trial $k$ in frequency $f$ and time bin $t$, and $N$ was the number of trials. The PPL was in [0 1], where 1 indicates high phase locking and 0 is no phase locking. Although within-area phase locking did not significantly differ between correct and wrong trials in the beta band (see Supplementary Information), we nevertheless used a trial shuffling procedure to remove any effect of within-area phase locking from the inter-area phase locking calculation. A shuffled distribution for each pair was calculated by randomly shuffling the pairing of trials for each site 1001 times. We then subtracted the mean of the shuffled distribution from the raw PPL value. In order to compare the conditions, we calculated Z-transformed PPL values by subtracting its value during baseline and dividing by the standard deviation across pairs for each condition. For visualization purposes (but not statistical comparisons), the time-frequency map was smoothed with a window of 2 Hz in frequency and 75 ms in time. Then, Z-transformed values were averaged across sites. PPL values in Fig. 2c–e were averaged across the delay period for the beta band.

The phase difference distributions were calculated similar to PPL, Eq. (2), except instead of absolute value the angle was calculated. For each pair the circular average was calculated over beta band frequencies and the delay period analysis window. The circular histogram of correct and wrong trials for Pref, In condition was plotted in Fig. 3a. Von Mises circularity tests were used to check the concentration of the distributions of the phase difference values of pairs (Fig. 3b). Concentration parameter, $\kappa$ values, between the correct and wrong trials were compared using a bootstrap test ($n = 1001$, each repetition using 63 LFP pairs selected randomly with replacement).

SPL measurements in Fig. 4a, b were performed on 221 spike-LFP pairs (IT units, FEF LFP), 182 spike-LFP pairs (IT units, IT LFP), 170 spike-LFP pairs (FEF units, FEF LFP), and 120 spike-LFP pairs (FEF units, IT LFP) in both monkeys. For each spike, we made a vector with the instantaneous phase at the spike time and an amplitude of one. The spikes from all trials were pooled together, and then we used the vector averaging method to calculate the SPL magnitude. In order to control for any effects of different numbers of spikes and trials between the two conditions, we considered a fixed window of 15 spikes and measured the SPL magnitude for the spikes of each window, and then took the mean across these windows[55].

In Fig. 5, PPL was calculated for single trials to split data into High and Low PPL trials. Single-trial PPL was calculated based on Eq. 2 across the delay window and baseline in one trial[68]. N was the number of time points in the window. We subtracted the PPL value of the baseline from the delay period PPL to remove the $1/f$ effect. Trials were ranked based on single-trial PPL values within each condition. We selected trials with PPL <33th percentile as Low and trials with PPL >66th percentile as High trials. Object discriminability was quantified using the receiver operating characteristic (ROC) analysis[69]. ROC analysis was performed on the distributions of neuronal firing rate. The areas under ROC curves (AUC) when comparing responses to different sample objects were used as an index of object discriminability, and were calculated as in previous studies[36]. Sometimes there were not enough trials of a given condition with High or Low PPL to compute an ROC (sample numbers are reported in the text for all statistics). For the timecourse of object discriminability of IT units, ROC was calculated for a jumping 300 ms window (10 ms steps). For Figs. 5a and S7a, we selected IT units with a discriminability of above chance level by permutation test ($n = 1001$; $p < 0.05$). Figures 5b, c and S7b, c were similar to Fig. 5a, but based on IT or FEF LFP power (instead of PPL). For Fig. 5d we calculated the modulation index as $(AUC_{High} - AUC_{Low})/(AUC_{High} + AUC_{Low})$. In Figs. 5f and S7d, we

calculated the Kendall correlation between the PPL difference (High - Low trials) and overall object discriminability, across sites. P-value was calculated by permutation test ($n = 1001$). In Fig. S5, we calculated spatial selectivity of FEF responses using the AUC for the In vs. Out conditions, using a 300-ms window beginning 50 ms after sample onset.

**Statistical analysis.** Statistical analyses were performed using the two-sided Wilcoxon's signed-rank test (for paired comparisons) or rank sum (for unpaired comparisons), unless otherwise specified. Statistical comparisons were performed across the population. For example, in comparing PPL for correct vs. wrong trials: for each LFP-LFP pair, PPL in each frequency band was calculated for correct trials, and for wrong trials. The Wilcoxon sign-rank test was then used to compare PPL values for correct trials vs. wrong trials across the population. Statistics for individual monkeys are reported in Table S1, and behavioral correlation statistics for all frequency bands and metrics are reported in Table S2. All p-values and effect sizes are reported up to 3 digits and values below 0.001 are reported as $p < 10^{-3}$. Effect size (Table S1) was calculated using Cohen's d.

**Reporting summary.** Further information on research design is available in the Nature Research Reporting Summary linked to this article.

## Data availability
Data are available from the authors upon request. A reporting summary for this Article is available as a Supplementary Information file. Source data are provided with this paper.

## Code availability
Code for analysis and reproduction of main conclusions is available online at https://github.com/FEF-IT-Object-Working-memory/source-code.

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

## Acknowledgements
The work was supported by NIH through EY026924, R01MH121435, and R01NS113073 to B.N., EY014924 to T.M., and EY014800 and an unrestricted grant from Research to Prevent Blindness, Inc. (New York, NY) to the Department of Ophthalmology and Visual Sciences, University of Utah. M.R.A.D was supported by Cognitive Sciences & Technologies Council (#2698).

## Author contributions
E.R., M.-R.A.D., B.N., K.C., and T.M. designed the experiments. E.R., M.-R.A.D., K.C., and B.N. performed the experiments. E.R., M.-R.A.D., Z.B., and B.N. analyzed the data. E.R., M.-R.A.D., K.C., T.M., and B.N. wrote the manuscript.

## Competing interests
The authors declare no competing interests.
