## [Peer Review File · Nature Communications]

Reviewers' comments:

Reviewer #1 (Remarks to the Author):

This is a potentially interesting manuscript examining the coordination between prefrontal and temporal cortex in a working memory task. I commend the authors for their dual recordings in IT and FEF in behaving monkey. However, my enthusiasm is diminished by the use of single-electrode recordings in each area rather than multi-electrode recording (which would have been more appropriate given the question asked – interareal coordination – and the analysis involved). As expected, mean firing rates and LFP power were poor predictors of behavioral performance (correct/wrong decisions). However, the phase locking of IT spikes with the beta-band LFPs in FEF predicted the correct behavioral responses. A second finding is that cells in IT had a greater object selectivity when IT spikes were phase locked to the FEF LFPs. These findings are interesting and can potentially provide evidence for the role of coordinated frontotemporal neural activity in the successful maintenance of visual information during working memory. Nonetheless, the results presented by the authors do not seem entirely consistent with the claims. There are numerous technical concerns and interpretation issues throughout the manuscript.

1. I am confused about the number of IT units reported in this study. They recorded from 58 IT sites for a total of 228 units, which means that on average they detected 4 units per site. Even for sharp electrodes this number is quite high and may suggest oversorting, which is a common problem in electrophysiology studies. How sure are they that the 4 units detected on each IT site (and possibly more on selected sites since 4 is the average) are different? They do not report any cross-correlation or auto-correlation analysis for the units on the same site to make sure they are not oversampling. This issue is particularly critical for the subsequent spike-LFP phase analysis (see below), which may introduce unwanted confounds.

2. Receptive fields (RFs) in IT and FEF are quite large. Perhaps I missed it, but don't see any statement mentioning that they recorded from cells/sites with overlapping RFs. How many units had overlapping or partially overlapping RFs? I assume that only a fraction of their recorded units satisfied this criterion. This is absolutely critical for the pairwise analysis (simultaneous FEF-IT recordings) reporting a link between WM and beta coupling, and more detail needs to be provided.

3. For the interareal PPL analysis in Figs. 2-3 it is critical to ensure that the data is coming from distinct units within the same area. How many pairs were used for each session? In Fig. 2c-e they report $n=45-50$, and I assume these are independent sessions pooled across monkeys. But I'd be skeptical of multiple spike-LFP pairs originating from the same sites (which could be corrupted by oversampling, see point 1 above). Do the results hold if they eliminate the excess units on each site? This is tricky, and could hurt the P value.

4. I'm confused about how the results are presented. Figs. 2-3 present results from the LFP-LFP and spike-LFP PPL analysis, but whether these analyses go in the same direction to prove that WM maintenance is consistent with more beta coherence is unclear. The presentation and writing need to be

much clearer to specify the differences between LFP-LFP and spike-LFP analyses. For instance, Fig. 3b shows that there is a small dip in SPL for wrong choices for FEF LFP – IT spike, but IT LFP – FEF spike and IT spike – IT spike analyses don't show differences. Why is this important? And where is the FEF spike – FEF spike analysis presented?

5. Fig. 2a shows a surprising (eerie) decrease in beta coupling for the wrong responses, but an increase for correct responses. Altogether, this amplifies the overall effect. How many sessions show this result? I find it surprising, but it is not discussed.

6. How can we make sure that behavioral choices are related exclusively to beta coupling in the WM interval, but not to the stimulus-locked coupling in other frequency bands? They need to analyze their data in a manner similar to that in Figs. 2-3 for the time intervals when stimuli are presented (in each condition). This will increase my confidence in the results.

7. In order to prove that the effects they see (Wr/Cr trials) are indeed due to WM and interareal beta coupling, other factors need to be ruled out. Two potential candidates are fixational eye position and movements and attention (but other factors may play a role too). They need to comprehensively analyze the fixational pattern to verify whether correct/wrong responses are related to differences in eye position (which may reflect different strategies employed in the Wr and Cr trials) or eye movements/velocity. Additionally, they need to rule out attention as a contributing factor, especially that attentional effects are strong in FEF. I don't see any gamma power difference between Cr/Wr trials, but how about other markers of attention in FEF and IT related to spiking activity?

8. The analysis in Fig. 4 (object coding in IT during beta PPL) is interesting, although I fear that the results may be contaminated by the oversorting issue mentioned on point 1. For instance, Fig. 4b reports a P value of 0.021 for 90 IT units, but the significance could be hurt if a large number of units would prove to be a duplicate of the same unit (due to oversorting). It seems to me that the main effect in the scatter plot (which seems quite small) is caused by the 8-10 extreme points above the diagonal. If some of those points were to disappear I can see this resulting in a statistically nonsignificant effect.

Reviewer #2 (Remarks to the Author):

The paper examined neuronal activity recorded simultaneously in the inferotemporal (IT) cortex in the FEF while monkeys compared two shapes separated by a brief delay, sample and target, and made a saccade to the shape presented during the sample. The authors focused on the behavioral relevance of neuronal activity in these two interconnected regions, the IT cortex implicated in shape processing and the FEF known for its saccade-related spatially selective activity. The goal was to compare the activity in the two areas and the interactions between them on correct and on error trials. The authors found differences in activity recorded during the delay on correct and error trials that were detectable only in the measure of the interactions between the two areas, the phase-locking of IT spikes and the beta LFP

frequency in FEF. This beta coupling between FEF and IT was revealed only on correct trials and its presence was correlated with improved object coding in IT. The authors concluded that the interactions between FEF and IT contribute to stimulus maintenance during the delay, providing strong evidence of the importance of coordination between prefrontal and object processing cortical neurons for successful task performance.

This is an important and timely study. The work is novel and technically challenging. The data are carefully analyzed, providing strong documentation in support of the conclusion that the interaction between FEF and IT have behavioral consequences.

The results are likely to be of interest to neuroscientists interested in working memory and in the neural basis of perceptual decisions.

I have a few comments and suggestions.

1. More attention should be devoted to the presentation of behavioral performance.

Since behavioral performance (correct/errors) guides this analysis it is important to consider potential sources of errors during the MTS task, which consists of several stages: sample (encoding), delay (retention), target (comparison). While it is reasonable to assume that the main source of errors during this task is “forgetting” the sample, it is also possible that on some trials the animal missed the 300ms sample or failed to distinguish the target from the remembered sample. This question could be addressed to some extent by testing monkeys at a short delay. If the main source of errors was forgetting the sample, it is likely that the performance would be better at shorter delays. I suspect that these behavioral data already exist in the lab and could be presented in the paper. This is an important question since the results were interpreted as evidence of the role of coordinated cortical activity for maintaining of sensory signals during the delay.

Does the delay length account for the difference in performance between the two animals (70 vs 77% correct)? How many trials (sessions) were used to compute % correct for each animal shown in Fig 1b?

Latencies in response to targets on correct and error trials could also provide useful insights into the nature of errors in each animal.

2. Given the difference in performance between the two monkeys, the recording data for individual animals should be presented for each analysis.

3. The rationale for limiting the analysis of delay activity in IT to the activity following the preferred stimulus is not clear.

4. In figure 1C the response to target on correct trials appear stronger than on errors. Is that a real effect?

5. On correct trials, beta coupling in the delay appears transient and absent at the very end of the delay.

Given the notion that “beta coupling predicts the maintenance of object identity within IT cortex” the implications of the absence of coupling at the end of the delay should be discussed.

6. The data points shown in scatterplots in Figs 2c-e represent average PPL across the entire delay. Given that beta coupling appears transient during the delay, it may be useful to show the effects for different periods in the delay.

Reviewer #3 (Remarks to the Author):

In this report, Rezayat and colleagues explore signals related to performance in a delayed match to sample (DMS) task by recording from prefrontal (FEF) and inferotemporal (IT) cortical areas in the monkey. These areas are known to be involved in spatial planning and object processing, respectively, and the authors ask how activity from each area alone, as well as coordinated activity between the areas, predict correct or wrong choices in the animals' matching task. In general, they find that information in each area alone, either in the form of spiking activity or slower field potentials, does not predict performance. By exploring the relationship between spiking activity in IT with the phase of signals in the beta frequency range in FEF, they find that the degree of phase locking does predict behavior. Further, object selective delay activity was enhanced when spikes were phase locked to FEF field potentials. Together, the results suggest that memory performance relies not just on neural responses in particular brain regions, but rather the coordination of these responses across regions.

General Comments

The basic premise and approach of this study is very compelling. Understanding how neural signals across distributed brain regions is undoubtedly one of the most important directions for neurophysiological studies related brain and behavior. That activity of neurons in specific brain areas is not easily related to performance in complex cognitive tasks certainly highlights the possibility that critical linking signals are only evident when activity from multiple, interacting, areas is observed simultaneously. Using that approach, in behaving monkeys, is the starting point for this study. I have some fairly significant concerns about the study, however, outlined below, mostly related to the strength of the data and their support for the authors' conclusions.

1. One obvious concern, which is not made entirely clear until the methods are scrutinized, is that the data from the two animals in this study come from two separate labs. In principle, this can actually be seen as a positive, as it provides some degree of generalization of the findings. The problem is that the methods are slightly different (quite different electrodes are used in the two sites for the IT recordings) and the systems for running the experiments are different. That alone is still not a major problem, but

without seeing the data broken down by animal in all the scatter plots, the fact that two animals were run in this study is not very compelling. We have come to accept 2 monkey studies, and this is again not my concern, but the authors really need more information about the data from the two animals beyond figure 1b (which simply shows a bar chart for overall performance). I realize Supp Fig 1 attempts to address this, but all the figures in the paper should include information where appropriate.

2. Conceptually, I think the authors need to discuss the relationship between their two findings and explain why together these are not in conflict with their suggestion that activity within a single area are poor predictors of behavior. Specifically, the claim is that phase locking of IT spikes (and LFPs) with FEF beta band activity predicts performance. Second, the authors assert increased phase locking of IT spikes is also associated with increased object selective persistent activity. The question I can't work out, then, is why then, increased object selective persistent activity doesn't predict behavioral performance? I could certainly be missing why this should not be so, but it seems to me that it's a question of signal to noise, wherein more trials should provide adequate data to show that indeed object selectivity for correct trials is enhanced for correct vs wrong trials. At least the authors should guide the reader through this question.

3. The link between the FEF selectivity and object selectivity is not made entirely clear. In particular, the authors establish the FEF selectivity by stimulation so they can put stimuli "in" or "out" of the cells' receptive fields. To this end, more details about the distribution of RFs should be provided, as this had a major impact on the task design and the potential location of stimuli. How consistent were these session to session? Also not clear was what was the prediction about the relationship between the FEF selectivity and the likelihood of observing phase locking and links to performance? Presumably there was no link between overall behavior and stimulus location, but we have virtually no analysis of behavior beyond correct/incorrect proportions. Much more could and should be done with the behavioral data if the key message is that neural activity in a single region doesn't predict behavior. Overall, I found the section describing the lack of significant predictions of behavioral performance from the single area firing rates and delay LFP power to seem a bit superficial (so as to move on to showing how other predictions were significant). Given the reliance on the null single area effects, the authors should really show how hard they looked to find these as opposed to providing a cursory analysis.

4. The main findings, shown in Figure 2a are certainly promising. I think the authors need to do a better job explaining why increased PPL between areas is not an indication of a more general form of task engagement (not specific to DMS performance). Further, the data in 2d and 2e are not very compelling. For 2d, the authors need to be a bit clearer about how effective stimuli were selected (from the methods this seems to have been by ear?).

5. I found the data shown in Figure 3 to be very interesting. From 3a I would like to really understand these data. Why, for example, does it seem like these two areas are generally in-phase, as the $\phi=180$ direction is underrepresented. Is this expected? Is this driven by the stimulus onset transients? In 2b I don't understand how the selective frequency bands were identified. It seems like differences in low and medium gamma were also different? Was a randomization procedure used to identify significant differences? Also, for the dependence on stimulus Pref for SPL difference, it's not clear that the number of spikes (which is of course different in these conditions) would not bias these estimates?

6. For the data shown in Figure 4, what's the relation between the sorting and performance (related to point 2, above).

7. More details about the eye movements made within the 1.5 dva should be provided, given that the data set relies so heavily on signals obtained from spatially selective FEF sites, where movements of the eyes on certain trials would certainly affect activity. Can, for example, raw eye position data along with stimulus location, predict behavior?

Minor Comments

1. Figure 1b: it's not really clear what we learn from this bar plot . One thing that seems apparent is that behavior between the two animals was actually significantly different? The number of sessions should be included here (and in the text)
2. Figure 1c and 1d: why is no different plot shown?
3. How the experimenters decide on the size of the stimulus set to use in the DMS task? Would you predict a smaller or larger set would affect the results of the study?
4. Figure 3 introduces the acronym SPL before the text describes spike-phase locking
5. Figure legend for 4 refers to "histograms" where it seems like there is only one histogram and the last sentence ("Same as (e) but for the Out condition.") seems out of place?
6. Check for misspelling of trial as "trail"

Reviewer #1 (Remarks to the Author):

This is a potentially interesting manuscript examining the coordination between prefrontal and temporal cortex in a working memory task. I commend the authors for their dual recordings in IT and FEF in behaving monkey. However, my enthusiasm is diminished by the use of single-electrode recordings in each area rather than multi-electrode recording (which would have been more appropriate given the question asked – interareal coordination – and the analysis involved).

We agree with the reviewer that multi-electrode recordings would have been preferable, as it would have likely improved the quality of the data and the robustness of the results, particularly in terms of the quantity of data per session. Indeed, if we were to begin another set of similar recordings from these areas, we would likely use the newest generation of multi-electrodes (e.g. Neuropixels), as they would likely require many fewer experimental sessions to achieve the significant effects that we observed with single electrodes, i.e. a lot fewer than the 92 recording sessions we performed. Nonetheless, as we demonstrate more clearly in the revised paper, the number of recording sessions we acquired with single electrodes was sufficient to obtain statistically reliable inter-area LFP power correlations, phase lag, phase locking, and spike-phase locking.

As expected, mean firing rates and LFP power were poor predictors of behavioral performance (correct/wrong decisions). However, the phase locking of IT spikes with the beta-band LFPs in FEF predicted the correct behavioral responses. A second finding is that cells in IT had a greater object selectivity when IT spikes were phase locked to the FEF LFPs. These findings are interesting and can potentially provide evidence for the role of coordinated frontotemporal neural activity in the successful maintenance of visual information during working memory. Nonetheless, the results presented by the authors do not seem entirely consistent with the claims. There are numerous technical concerns and interpretation issues throughout the manuscript.

We have edited the text and figures to make our findings clearer and added multiple control experiments and technical details of the analysis, and we believe that our responses below will reassure the reviewer about these technical and interpretation issues.

1. I am confused about the number of IT units reported in this study. They recorded from 58 IT sites for a total of 228 units, which means that on average they detected 4 units per site. Even for sharp electrodes this number is quite high and may suggest oversorting, which is a common problem in electrophysiology studies. How sure are they that the 4 units detected on each IT site (and possibly more on selected sites since 4 is the average) are different? They do not report any cross-correlation or auto-correlation analysis for the units on the same site to make sure they are not oversampling. This issue is particularly critical for the subsequent spike-LFP phase analysis (see below), which may introduce unwanted confounds.

We understand the reviewer's confusion on this issue given that the numbers of units and sessions were not described clearly enough in the previous version of the paper. As we clarify in the revised paper, we report single-neuron data from 92 simultaneous FEF/IT recording sessions (6 additional recording sessions were added to the revised paper, the previous version included 86 sessions). In total, there are 235 units from these 92 recording sessions in IT, and 170 units in FEF. The average number of units per site is thus 2.5 in IT and 1.8 in FEF, which are both within the normal range for single electrode recordings (Pedreira et al. *J Neurosci Methods* 2012, Harris et al. *Nat Neurosci* 2016, Camuñas-Mesa et al. *Neural Comput.* 2013).

In the revised paper, we have added a more rigorous assessment of the quality of the waveform sorting, namely an SVM analysis, which has been used previously by us and others (Zirnsak et al. *Nature* 2014 and Merrikhi et al. *Nat Comm* 2017; Barnett et al. *J Neurosci Methods* 2016). This formal approach increases the confidence in sorted spikes well above the more traditional approach, which has largely been informal and/or unspecified. We have added this analysis to the Methods section and Fig. S1g: “*The overall high performance of the classifier indicates that the spikes of each sorted cluster is well isolated from each other cluster ($\Delta\text{PerfIT} = 98.80\% \pm 2.50$, $n = 301$ pairs, $p < 10^{-50}$, $\Delta\text{PerfFEF} = 98.67\% \pm 4.018$, $n = 105$ pairs, $p < 10^{-18}$ compared to chance level, i.e. 50%).*”

2. Receptive fields (RFs) in IT and FEF are quite large. Perhaps I missed it, but don't see any statement mentioning that they recorded from cells/sites with overlapping RFs. How many units had overlapping or partially overlapping RFs? I assume that only a fraction of their recorded units satisfied this criterion. This is absolutely critical for the pairwise analysis (simultaneous FEF-IT recordings) reporting a link between WM and beta coupling, and more detail needs to be provided.

We thank the reviewer for requesting this clarification, as it should not be taken for granted. Indeed, all of the FEF and IT units included in the analysis had overlapping RFs. This was by design, because as the reviewer notes this overlap is critical for these analyses: sample stimuli were placed in the estimated FEF RF for In trials, and a visual response to a preferred stimulus was a criteria for inclusion for IT units (that IT and FEF units do both respond to stimuli at the In location can be seen in the PSTHs in Fig. 1c). Detailed mapping of RFs was not conducted; as the reviewer notes, IT receptive fields are quite large, encompassing much of the contralateral and some of the ipsilateral hemifield (Gross et al *Science* 1969), and in our study included all of the low-eccentricity (<13dva) contralateral visual hemifield where stimuli were placed for these recordings. When the stimulus appeared in the ipsilateral hemifield, it was always outside the FEF RF, but sometimes still evoked a visual response in IT (in general, neural modulation was only seen for the In condition, when the sample appeared in both the IT and FEF RF, e.g. Fig. 5d). We now provide more details on the distribution of FEF RFs in Fig. S1, and we have clarified this point about the overlap between IT and FEF RFs in the main text and in the methods section: “*We recorded spiking activity and LFP signals*

simultaneously from FEF and IT sites with overlapping receptive fields (RFs) in two monkeys (M1, M2) using two single electrodes (92 sessions; 239 IT units, 69 LFP sites; 170 FEF units, 92 LFP sites)(see Methods). All of the FEF and IT units in the analysis had overlapping RFs, both responding to visual stimuli in the FEF RF.”, “By design, all of the FEF and IT units recorded had overlapping RFs. IT RFs are large⁵⁸, and typically include the low eccentricity (<13 dva) contralateral visual hemifield in which all our FEF RFs were centered. Both FEF and IT neurons responded to stimuli placed in the estimated FEF RF (Fig. 1c).”.

3. For the interareal PPL analysis in Figs. 2-3 it is critical to ensure that the data is coming from distinct units within the same area. How many pairs were used for each session? In Fig. 2c-e they report n=45-50, and I assume these are independent sessions pooled across monkeys. But I’d be skeptical of multiple spike-LFP pairs originating from the same sites (which could be corrupted by oversampling, see point 1 above). Do the results hold if they eliminate the excess units on each site? This is tricky, and could hurt the P value.

Note that the PPL calculation in figure 2 does not use spiking data, only LFPs. We have also added an SVM analysis to verify the quality of our spike-sorting (details above). Nonetheless, we agree that it is worthwhile to address the possibility of oversampling single neurons, and thus for data in figure 4, we re-did the analysis as suggested, using only a single randomly selected unit from each recording session (n = 79 spike-LFP pairs). Using only 1 single neuron per session of course provides the most conservative test of the SPL effect. We describe this additional test in the text of the Supplementary Information: “For the difference in SPL for correct vs. wrong trials (Fig. 4a), we measured the distribution of the median difference in correct vs. wrong SPL for 1000 random subsamples of 79 spike-LFP pairs, each with a maximum of one unit per recording session. For > 99.5% of the subsamples the SPL for correct was greater than wrong, and the mean of this distribution was significantly greater than

zero ($\Delta SPL_{Cr-Wr} = 0.008 \pm 0.000$, $p < 10^{-10}$, $n = 1000$).”

Control for Fig. 4a: Histogram shows the distribution of the median difference in correct vs. wrong SPL for 1000 subsamples (each with 79 spike-LFP pairs, a maximum of one unit per recording session)

4. I'm confused about how the results are presented. Figs. 2-3 present results from the LFP-LFP and spike-LFP PPL analysis, but whether these analyses go in the same direction to prove that WM maintenance is consistent with more beta coherence is unclear. The presentation and writing need to be much clearer to specify the differences between LFP-LFP and spike-LFP analyses.

To reduce the confusion, we have now separated the PPL, phase difference, and SPL results into different figures, and also added further section headers to reduce confusion about the type of data involved; figure 2 is all LFP-LFP phase locking results, figure 3 shows LFP phase differences, and figure 4 shows spike-phase locking. We believe this change will make the results clearer.

For instance, Fig. 3b shows that there is a small dip in SPL for wrong choices for FEF LFP – IT spike, but IT LFP – FEF spike and IT spike – IT spike analyses don't show differences. Why is this important?

Our interpretation of this result is that it reflects a modulatory influence of FEF LFPs in the beta band on IT spiking activity. Importantly, the absence of such an effect in the other spike-LFP combinations of the SPL analyses suggests that the first effect is not just a global increase in all measures of beta-band coupling, but is specific to a directional interaction between these areas. We have clarified this interpretation by adding this sentence to the main text: *"The inter-area specificity of these signatures, and the directionality of the SPL effect (FEF LFP to IT spikes only), suggests a model in which FEF beta activity modulates IT spiking."*

And where is the FEF spike – FEF spike analysis presented?

The FEF spike- FEF LFP analysis has been added in Fig. 4c.

5. Fig. 2a shows a surprising (eerie) decrease in beta coupling for the wrong responses, but an increase for correct responses. Altogether, this amplifies the overall effect. How many sessions show this result? I find it surprising, but it is not discussed.

We thank the reviewer for highlighting this. 59% of sessions show an increase in delay period coupling for correct trials, and 57% of sessions show a decrease on wrong trials, and 34% of sessions show both. We have added statistics on this decrease to the main text: *"Importantly, we observed a significant enhancement in beta band PPL during the delay period compared to baseline ($\Delta PPL = 0.267 \pm 0.108$, $p = 0.035$, $n = 69$; 37 of 69 pairs showed a significant increase), indicating a more consistent phase relationship between beta LFPs in FEF and IT during successful memory maintenance, and a significant decrease in wrong trials ($\Delta PPL = -0.211 \pm 0.095$, $p = 0.028$, $n = 63$; 39 of 69 pairs showed a significant decrease)."*

6. How can we make sure that behavioral choices are related exclusively to beta coupling in the WM interval, but not to the stimulus-locked coupling in other frequency bands? They need to analyze their data in a manner similar to that in Figs. 2-3 for the

time intervals when stimuli are presented (in each condition). This will increase my confidence in the results.

Importantly, the results in the main text are shuffle-corrected (randomly pairing FEF and IT trials within each condition, and subtracting the mean shuffled value), and so any phase locking within an area as result of stimulus-locked coupling is removed. In fig S3a the results without shuffle correction are shown, and the effects in various frequency bands are now reported in the supplementary text: *“PPL statistics and data presented in figure 2 and the main text used a shuffling procedure to remove any effect of within-area phase locking (see Methods). Without this shuffling, there was still significantly higher beta band PPL on correct vs. wrong trials (Fig. S3a right; $\Delta PPL = 0.347 \pm 0.100$, $p = 0.002$, $n = 63$). PPL values, not shuffle-corrected, increased in the alpha, and beta bands during the visual period vs. baseline ($\Delta PPL_{\alpha} = 0.218 \pm 0.084$, $p = 0.037$, $\Delta PPL_{\beta} = 0.279 \pm 0.112$, $p = 0.025$, $n = 69$). During the visual period, only the beta band PPL was object selective ($\Delta PPL = 0.457 \pm 0.172$, $p = 0.011$, $n = 69$) and location selective ($\Delta PPL = 0.546 \pm 0.158$, $p < 0.001$, $n = 69$). During the visual period there was significantly higher beta band PPL on correct vs. wrong trials (Fig. S3a right; $\Delta PPL = 0.584 \pm 0.174$, $p = 0.002$, $n = 63$).”*

We also report shuffle-corrected values for the visual period in various bands in the Supplementary Information: *“PPL values, shuffle-corrected, increased in the alpha, and beta bands during the visual period vs. baseline ($\Delta PPL_{\alpha} = 0.218 \pm 0.084$, $p = 0.037$, $\Delta PPL_{\beta} = 0.279 \pm 0.112$, $p = 0.025$, $n = 69$). During the visual period, only the beta band PPL was object selective ($\Delta PPL = 0.457 \pm 0.172$, $p = 0.011$, $n = 69$) and location selective ($\Delta PPL = 0.546 \pm 0.158$, $p < 0.001$, $n = 69$). During the visual period there was significantly higher beta band PPL on correct vs. wrong trials (Fig. S3a right; $\Delta PPL = 0.453 \pm 0.128$, $p = 0.001$, $n = 63$).”*

7. In order to prove that the effects they see (Wr/Cr trials) are indeed due to WM and interareal beta coupling, other factors need to be ruled out. Two potential candidates are fixational eye position and movements and attention (but other factors may play a role too). They need to comprehensively analyze the fixational pattern to verify whether correct/wrong responses are related to differences in eye position (which may reflect different strategies employed in the Wr and Cr trials) or eye movements/velocity.

We have now added an analysis of eye position during the task, and the rate of fixational microsaccades (>0.1 and <1 dva), to Fig. S1e-f and the supplementary text. Although the eye position is biased toward the sample location, there is no difference in either eye position or microsaccade rate for correct vs. wrong trials.

Additionally, they need to rule out attention as a contributing factor, especially that attentional effects are strong in FEF. I don't see any gamma power difference between Cr/Wr trials, but how about other markers of attention in FEF and IT related to spiking activity?

The effects of attention on neuronal activity, including within FEF and IT, are generally observed when a particular stimulus or location is cued or otherwise selected over other stimuli or locations (e.g. Zhang et al *PNAS* 2011, Reynolds et al *J Neuro* 1999, Zhou & Thompson *Vis Res* 2009). Such circumstances differ critically from working memory tasks like ours, where only a single stimulus is presented and then remembered over a

delay period. Attention effects, either psychophysical or neurophysiological, generally require comparisons of detection or neural responses to distracting non-target stimuli (e.g. Carrasco *Vis Res* 2011, Desimone & Duncan *Ann Rev Neuro* 1995). Although it is possible that attention to the single, sample stimulus differed between correct and wrong trials, the fact that there were no distracters suggests that variation in attention to the sample was likely very small, as there was minimal attentional load (e.g. Boudreau et al. *J Neurophys* 2006). Consistent with that fact, we observed no differences in IT responses to the sample objects between correct and wrong trials ($\Delta\text{NFR} = 0.004 \pm 0.006$, $p=0.524$, $n=232$), which would be expected if attentional modulation during the sample period was a contributing factor (Moran & Desimone *Science* 1985; Sheinberg & Logothetis *J Neuro* 2001). Nevertheless, a key result of the paper is the robust difference in the synchrony of IT spikes with the FEF LFPs between correct and wrong trials during the delay period.

In contrast to the neural correlates of attention, activity in the memory delay period is traditionally believed to be the correlate of working memory (Fuster & Alexander *Science* 1971, Funahashi et al *J Neurophys* 1989). During the delay period, sensory stimuli are absent and information is only available in memory. Nonetheless, the issue of whether attention and working memory can truly be dissociated is a well-known and challenging question (e.g., Awh and Jonides *Trends Cog Sci* 2001). Indeed, according to some theories, attention consists of nothing more than directing working memory to a sensory stimulus, or to an internal representation of stimuli when they are no longer present (e.g. Knudsen *Ann Rev Neuro* 2007, Chun *Neuropsych* 2011, Griffin and Nobre *J Cog Neuro* 2003). We are inclined to subscribe to this view of the relationship between WM and attention, i.e. that they are not necessarily dissociable in spite of the clear differences between our task and those traditionally used to demonstrate attention effects. It is possible that attention may indeed be contributing to memory maintenance, not just the selection of cued stimuli, and that the observed interareal beta coupling is a signature of that attention operating during memory maintenance. We have attempted to clarify our understanding of this potential role of attention during the WM task by adding the following text to the discussion section: *“In our data, coherence between IT and the FEF could reflect the deployment of spatial attention for feature binding during object memory maintenance^{40,47-49}. ... These results suggest that perhaps, similar to the case of spatial attention (or indeed due to a role of spatial attention in memory maintenance^{44,51,52}), a non-selective signal highlighting relevant information is sufficient to boost object-selective spiking activity in IT.”*

Although we do not believe the effects of attention necessarily can or should be completely eliminated, and indeed virtually no studies of WM clearly eliminate those possible effects, we can nonetheless control for potential intermediate neural effects. As the reviewers note, there was no gamma power difference between correct and wrong trials (nor is there a power difference in any other band; see Table S1). There is a slight difference in firing rate in FEF between correct and wrong trials, but this difference was controlled for in the SPL analysis- we have clarified this point in the Methods section: *“In order to control for any effects of different numbers of spikes and trials between the two conditions, we considered a fixed window of 15 spikes and measured the SPL magnitude for the spikes of each window, and then took the mean across these windows⁴⁷.”*

8. The analysis in Fig. 4 (object coding in IT during beta PPL) is interesting, although I

fear that the results may be contaminated by the oversorting issue mentioned on point 1. For instance, Fig. 4b reports a P value of 0.021 for 90 IT units, but the significance could be hurt if a large number of units would prove to be a duplicate of the same unit (due to oversorting).

We have addressed the reviewer's concerns regarding oversorting in point 1 above. However, to further reassure the reviewer that the results do not rely on having multiple units from each site, we also re-did the comparison of the change in object selectivity for High vs. Low PPL trials (currently reported in Fig. 5e) using only a single randomly selected spiking unit from each recording session ($n = 40$ units). We describe this additional test in the text of the Supplementary Information: "For the modulation of object selectivity on High vs. Low PPL trials (Fig. 5e), we calculated the distribution of the median difference in modulation index (modulation of object discriminability for High vs. Low PPL trials) for In vs. Out, for 1000 subsamples (each with 40 units, a maximum of one unit per recording session). For 100% of the subsamples the modulation index was greater for In vs. Out trials, and the mean of this distribution was significantly greater than zero ($\Delta MI_{In-Out} = 0.090 \pm 0.000$, $p < 10^{-10}$, $n = 1000$)."

Control for Fig. 5e: Histogram shows the distribution of the median difference in modulation index (modulation of object discriminability for High vs. Low PPL trials) for In vs. Out, for 1000 subsamples (each with 40 units, a maximum of one unit per recording session).

It seems to me that the main effect in the scatter plot (which seems quite small) is caused by the 8-10 extreme points above the diagonal. If some of those points were to disappear I can see this resulting in a statistically nonsignificant effect.

Note that the statistical comparison for previous figure 4b (the plot has been removed from the revised paper, but the statistic remains in the text) uses a nonparametric test (Wilcoxon signed-rank test), in which more extreme values do not contribute more to driving the p-value.

Reviewer #2 (Remarks to the Author):

The paper examined neuronal activity recorded simultaneously in the inferotemporal (IT) cortex in the FEF while monkeys compared two shapes separated by a brief delay, sample and target, and made a saccade to the shape presented during the sample. The

authors focused on the behavioral relevance of neuronal activity in these two interconnected regions, the IT cortex implicated in shape processing and the FEF known for its saccade-related spatially selective activity. The goal was to compare the activity in the two areas and the interactions between them on correct and on error trials. The authors found differences in activity recorded during the delay on correct and error trials that were detectable only in the measure of the interactions between the two areas, the phase-locking of IT spikes and the beta LFP frequency in FEF. This beta coupling between FEF and IT was revealed only on correct trials and its presence was correlated with improved object coding in IT. The authors concluded that the interactions between FEF and IT contribute to stimulus maintenance during the delay, providing strong evidence of the importance of coordination between prefrontal and object processing cortical neurons for successful task performance.

This is an important and timely study. The work is novel and technically challenging. The data are carefully analyzed, providing strong documentation in support of the conclusion that the interaction between FEF and IT have behavioral consequences. The results are likely to be of interest to neuroscientists interested in working memory and in the neural basis of perceptual decisions. I have a few comments and suggestions.

1. More attention should be devoted to the presentation of behavioral performance. Since behavioral performance (correct/errors) guides this analysis it is important to consider potential sources of errors during the MTS task, which consists of several stages: sample (encoding), delay (retention), target (comparison). While it is reasonable to assume that the main source of errors during this task is “forgetting” the sample, it is also possible that on some trials the animal missed the 300ms sample or failed to distinguish the target from the remembered sample. This question could be addressed to some extent by testing monkeys at a short delay. If the main source of errors was forgetting the sample, it is likely that the performance would be better at shorter delays. I suspect that these behavioral data already exist in the lab and could be presented in the paper. This is an important question since the results were interpreted as evidence of the role of coordinated cortical activity for maintaining of sensory signals during the delay.

We agree that the source of errors could be more than just “forgetting”. In principle, as the reviewer points out, there are at least three different periods when errors can occur in the MTS task: during encoding, maintenance, or target selection. As we now note in the methods, in designing the task, we intentionally chose a long sample duration (300 ms) in hopes of minimizing encoding errors (estimated 50 ms/sample needed for encoding each item, Vogel et al *J Exp Psych* 2006). Consistent with this goal, there was no difference in firing rate between correct and wrong trials during the sample period in the final stage of visual processing (IT) ($\Delta\text{NFR} = 0.004 \pm 0.006$, $p = 0.524$, $n = 232$), which might be expected if encoding were a major source of errors.

However, we wish to acknowledge that we cannot definitively ascribe errors solely to failures of memory maintenance. We have added a paragraph addressing the multiple potential sources of error to our discussion section: *“It is important to note that multiple*

sources of errors are possible during the object DMS task. Encoding errors could result from lapses in attention during the sample period, or if attention was incorrectly directed to the alternative sample location. During the delay period, there could be failures of maintenance¹⁹, or interference from samples remembered during previous trials²⁰. Effects of recent reward history, on the other hand, presumably influence choice during the target selection stage²¹. Errors in target selection could arise due to impulsivity or selection biases, either based on recent reward history or more persistent biases in target location or object identity. Despite this heterogenous mix of sources of errors, we were nevertheless able to identify interareal signatures that correlated with memory performance during the delay period. Observed differences between correct and wrong trials during the delay period may either reflect failures in memory, or failures to encode the stimulus. Clearly, if a sample is not encoded, it cannot be remembered, and thus error trials caused by failures of encoding should exhibit no memory-related activity. Errors in target selection following successful encoding and memory maintenance should result in mislabeling of trials with sufficient memory activity as 'wrong' trials, weakening the strength of behavioral correlations. Nevertheless, we observe significant behavioral correlations for neural signatures during the delay period, suggesting that either the magnitude of the effects we observed would be stronger when limiting analysis only to trials with failures of memory, or that there were comparatively few target selection errors."

Short-delay data would have provided an elegant control for both encoding and target selection errors. Unfortunately, we do not have short-delay data, nor are the animals currently available for further data collection.

Does the delay length account for the difference in performance between the two animals (70 vs 77% correct)? How many trials (sessions) were used to compute % correct for each animal shown in Fig 1b? Latencies in response to targets on correct and error trials could also provide useful insights into the nature of errors in each animal.

This is a good question, and we have added details on some of the behavioral differences between animals. The length of the delay period was actually the same for both animals (1 second). There were 51 sessions for Monkey 1 and 41 sessions for Monkey 2. We have added data on the reaction times for both monkeys in correct and error trials (Fig. S1a-b). Perhaps surprisingly, the animal with lower performance had slower reaction times; for this animal, correct trials had longer RTs, consistent with a speed/accuracy tradeoff. Monkey 1 performed more sessions at greater eccentricities (Fig. S1c), which could contribute to (and potentially explain) the overall lower performance. We have added these additional behavioral analyses to the Supplementary Information.

2. Given the difference in performance between the two monkeys, the recording data for individual animals should be presented for each analysis.

Some of the potential sources of the performance difference are discussed in the response to point 1 above. To show that effects are consistent between animals, we have updated all figures to show which data points came from each monkey, as well as providing statistics for individual monkeys in Table S1.

3. The rationale for limiting the analysis of delay activity in IT to the activity following the preferred stimulus is not clear.

We have clarified the rationale for focusing more on the responses to preferred stimuli. The rationale is that non-preferred stimuli evoke weaker visual responses, and consequently weaker delay activity, as we would expect (Fuster & Jervey *Science* 1981, Miyashita & Chang *Nature* 1988, Fuster *J Neurophys* 1990). We have added the following rationale to the text: “As in previous studies¹⁴⁻¹⁶, we focused on visual stimuli that evoked the strongest visual responses in IT, as they are more likely to be involved in maintaining the memory of those stimuli. Therefore, we analyzed the Pref condition for IT, and the sample In condition for the FEF. For analyses involving both FEF and IT, we considered the Pref In condition, unless otherwise specified.”

Note, however, that in Figure 5, in which object selectivity is measured by comparing preferred and nonpreferred responses, we do in fact make use of the nonpreferred data.

4. In figure 1C the response to target on correct trials appear stronger than on errors. Is that a real effect?

The split in target responses occurs after eye movement onset (average reaction times ~150ms, see Fig. S1a). In IT the correct trials have a higher firing rate ($\Delta\text{NFR} = 0.015 \pm 0.004$, $p < 10^{-4}$, $n = 232$). Although we generally don't try to interpret data after the eye is moving, in this case there is a pretty clear explanation: for IT, Fig. 1C only shows trials on which the preferred object was the correct choice, so this correct/wrong difference corresponds to selecting and foveating the preferred vs. other object; the same effect can be seen in the different responses to Pref/Inter/NPref objects during the target period for only correct trials in the left panel of Fig. 1C (wrong trials on the right will be a mix of saccades to Inter and NPref objects).

In FEF, the difference in firing rate in the target period is not statistically significant ($\Delta\text{NFR} = -0.004 \pm 0.004$, $p = 0.241$, $n = 170$).

5. On correct trials, beta coupling in the delay appears transient and absent at the very end of the delay. Given the notion that “beta coupling predicts the maintenance of object identity within IT cortex” the implications of the absence of coupling at the end of the delay should be discussed.

With additional recording sessions added to the revised paper, this does not appear to be the case. We have added additional analysis measuring object and location selectivity of PPL over the course of the delay period, and it persists until the end of the delay (Fig. S3c&d).

6. The data points shown in scatterplots in Figs 2c-e represent average PPL across the entire delay. Given that beta coupling appears transient during the delay, it may be useful to show the effects for different periods in the delay.

We have added additional analysis of object and location selectivity of PPL over the course of the delay period, and it persists until the end of the delay (Fig. S3c&d).

Reviewer #3 (Remarks to the Author):

In this report, Rezayat and colleagues explore signals related to performance in a delayed match to sample (DMS) task by recording from prefrontal (FEF) and inferotemporal (IT) cortical areas in the monkey. These areas are known to be involved in spatial planning and object processing, respectively, and the authors ask how activity from each area alone, as well as coordinated activity between the areas, predict correct or wrong choices in the animals' matching task. In general, they find that information in each area alone, either in the form of spiking activity or slower field potentials, does not predict performance. By exploring the relationship between spiking activity in IT with the phase of signals in the beta frequency range in FEF, they find that the degree of phase locking does predict behavior. Further, object selective delay activity was enhanced when spikes were phase locked to FEF field potentials. Together, the results suggest that memory performance relies not just on neural responses in particular brain regions, but rather the coordination of these responses across regions.

General Comments

The basic premise and approach of this study is very compelling. Understanding how neural signals across distributed brain regions is undoubtedly one of the most important directions for neurophysiological studies related brain and behavior. That activity of neurons in specific brain areas is not easily related to performance in complex cognitive tasks certainly highlights the possibility that critical linking signals are only evident when activity from multiple, interacting, areas is observed simultaneously. Using that approach, in behaving monkeys, is the starting point for this study. I have some fairly significant concerns about the study, however, outlined below, mostly related to the strength of the data and their support for the authors' conclusions.

1. One obvious concern, which is not made entirely clear until the methods are scrutinized, is that the data from the two animals in this study come from two separate labs. In principle, this can actually be seen as a positive, as it provides some degree of generalization of the findings. The problem is that the methods are slightly different (quite different electrodes are used in the two sites for the IT recordings) and the systems for running the experiments are different. That alone is still not a major problem, but without seeing the data broken down by animal in all the scatter plots, the fact that two animals were run in this study is not very compelling. We have come to accept 2 monkey studies, and this is again not my concern, but the authors really need more information about the data from the two animals beyond figure 1b (which simply shows a bar chart for overall performance). I realize Supp Fig 1 attempts to address this, but all the figures in the paper should include information where appropriate.

We agree with the reviewer on both counts: it is a positive to observe similar effects in different laboratories and different animals, but it is also important to provide details about the extent to which the results are indeed similar. We have therefore updated all

figures to show which data points come from each monkey. In addition, we have added separate statistics from the two animals to the Supplementary Information (Table S1) and additional information on the behavioral performance of both animals (Fig. S1a-e). As can be seen, the neural results are remarkably similar between animals/labs across the range of measurements, and in no case were the direction of effects different between the two.

2. Conceptually, I think the authors need to discuss the relationship between their two findings and explain why together these are not in conflict with their suggestion that activity within a single area are poor predictors of behavior. Specifically, the claim is that phase locking of IT spikes (and LFPs) with FEF beta band activity predicts performance. Second, the authors assert increased phase locking of IT spikes is also associated with increased object selective persistent activity.

Note that it is the *phase-phase locking*, not spike-phase locking, which is associated with increased object selectivity (Fig. 5). However, this distinction does not invalidate the reviewer's point.

The question I can't work out, then, is why then, increased object selective persistent activity doesn't predict behavioral performance? I could certainly be missing why this should not be so, but it seems to me that it's a question of signal to noise, wherein more trials should provide adequate data to show that indeed object selectivity for correct trials is enhanced for correct vs wrong trials. At least the authors should guide the reader through this question.

It is certainly possible that with sufficient data, IT spiking activity would be significantly correlated with behavior. The point we wish to make is not that no such relationship between spiking activity and behavior exists (and we report some behavioral correlation for FEF firing rates), but rather that the inter-areal coupling is a more sensitive measure- and, in our opinion, therefore more likely to reflect the mechanisms underlying successful maintenance. We have added this sentence to the discussion to clarify this point: *"Because oscillatory coherence predicted object selectivity in IT (Fig. 5), it is likely that with sufficient data IT object-selective activity would also be correlated with performance (as FEF activity was); however, these data show that the inter-area signatures are a more robust indicator of neural mechanisms related to maintenance."*

3. The link between the FEF selectivity and object selectivity is not made entirely clear. In particular, the authors establish the FEF selectivity by stimulation so they can put stimuli "in" or "out" of the cells' receptive fields. To this end, more details about the distribution of RFs should be provided, as this had a major impact on the task design and the potential location of stimuli. How consistent were these session to session?

The full distribution of estimated FEF RF positions (which determined where the samples were placed) is shown in the new Fig. S1d. We have also added a plot showing performance as a function of sample eccentricity (typically in the 5-13 dva range; Fig. S1c). In general, there was a negative relationship between sample eccentricity and performance (see Fig. S1c)- which makes sense given that stimuli were not scaled based on eccentricity.

Also not clear was what was the prediction about the relationship between the FEF selectivity and the likelihood of observing phase locking and links to performance?

As the reviewer suggests, we looked for a relationship between the magnitude of FEF selectivity for In vs. Out and the change in PPL with performance. Indeed, we found a positive correlation between the strength of FEF spatial selectivity and the change in PPL for correct vs. wrong trials ($r = 0.220$, $p < 10^{-5}$, $n = 136$; Fig. S4). We have added this result to the Supplementary Information (Fig. S4). We thank the reviewer for this suggestion.

Presumably there was no link between overall behavior and stimulus location, but we have virtually no analysis of behavior beyond correct/incorrect proportions. Much more could and should be done with the behavioral data if the key message is that neural activity in a single region doesn't predict behavior.

Additional analysis looking at the performance at different sample locations has been added in Fig. S1 and the Supplementary Information, along with an analysis of eye position and microsaccades. As discussed above, there are indeed multiple sources of error possible in this task, including biases based on sample or target location, or reward history effects between trials; however, any error which affects encoding/maintenance should alter neural signatures during the delay, and any error during the target period should only weaken our behavioral correlations. We have added a discussion of possible sources of error to the text: *“It is important to note that multiple sources of errors are possible during the object DMS task. Encoding errors could result from lapses in attention during the sample period, or if attention was incorrectly directed to the alternative sample location. During the delay period, there could be failures of maintenance¹⁹, or interference from samples remembered during previous trials²⁰. Effects of recent reward history, on the other hand, presumably influence choice during the target selection stage²¹. Errors in target selection could arise due to impulsivity or selection biases, either based on recent reward history or more persistent biases in target location or object identity. Despite this heterogenous mix of sources of errors, we were nevertheless able to identify interareal signatures that correlated with memory performance during the delay period. Observed differences between correct and wrong trials during the delay period may either reflect failures in memory, or failures to encode the stimulus. Clearly, if a sample is not encoded, it cannot be remembered, and thus error trials caused by failures of encoding should exhibit no memory-related activity. Errors in target selection following successful encoding and memory maintenance should result in mislabeling of trials with sufficient memory activity as ‘wrong’ trials, weakening the strength of behavioral correlations. Nevertheless, we observe significant behavioral correlations for neural signatures during the delay period, suggesting that either the magnitude of the effects we observed would be stronger when limiting analysis only to trials with failures of memory, or that there were comparatively few target selection errors.”*

Overall, I found the section describing the lack of significant predictions of behavioral performance from the single area firing rates and delay LFP power to seem a bit superficial (so as to move on to showing how other predictions were significant). Given the reliance on the null single area effects, the authors should really show how hard they looked to find these as opposed to providing a cursory analysis.

Additional details on behavioral correlations of firing rate and LFP are now provided in Figure S2, and additional details on within-area phase locking are also provided in the Supplementary text. The gamma frequency band was divided in two (based on visual inspection of plots) to increase the chances of finding behavioral correlations.

As mentioned above, the point we wish to make is not that there is *absolutely no* detectable relationship between spiking activity and behavior—indeed, we report a correlation between FEF spiking and performance. Rather, the inter-areal measures are a more robust indicator of performance. We have added a sentence to the discussion clarifying this point.

4. The main findings, shown in Figure 2a are certainly promising. I think the authors need to do a better job explaining why increased PPL between areas is not an indication of a more general form of task engagement (not specific to DMS performance).

Differences in task engagement should be expected to result in nonspecific differences in activity within and between areas. In contrast, the effects we report are quite specific to inter-areal coordination and to the delay period. For example, related to the discussion of attentional effects above, the fact that the firing rate in IT was no different between correct and wrong trials suggests that task engagement was not significantly different between these trial types. Nevertheless, PPL and SPL were different between correct and wrong trials.

Further, the data in 2d and 2e are not very compelling. For 2d, the authors need to be a bit clearer about how effective stimuli were selected (from the methods this seems to have been by ear?).

Prior to DMS recordings, effective stimuli for the IT unit were identified from a group of 44 potential stimuli based on multi-unit responses during an RSVP task. A combination of audible and computerized analysis were used to identify effective and ineffective stimuli. Certainly with only 44 potential stimuli we make no claim of having identified near-optimal stimuli for these IT units; however, the selected stimuli were adequate to drive the IT neurons, as can be seen based on their visual responses during the DMS task. The difference in IT responses to experimenter-selected effective and ineffective stimuli can be seen in the PSTHs for preferred vs. non-preferred stimuli in Fig. 1c (statistics on object selectivity of IT responses are in the Supplementary Information).

To give a fuller picture of how these values behave over the course of the task, we have added a timecourse of these values to the supplemental information as requested by another reviewer (Fig. S3b&d). Also note that more data have been added to the analysis in 2d and 2e, improving the p-values (from .009 and .025 to .001 and .016).

5. I found the data shown in Figure 3 to be very interesting. From 3a I would like to really understand these data. Why, for example, does it seem like these two areas are generally in-phase, as the $\phi=180$ direction is underrepresented. Is this expected? Is this driven by the stimulus onset transients?

It is true that these areas are generally in-phase. Based on the reviewer's question, we also evaluated the phase relationship during the fixation (prior to stimulus) and visual periods, and have added these results in new figure 3b. The reliability of the phase relationship across sessions is greater during the delay than the visual period for correct trials, suggesting that the visual onset transient is not the primary driver of the phase relationship.

In 2b I don't understand how the selective frequency bands were identified. It seems like differences in low and medium gamma were also different? Was a randomization procedure used to identify significant differences?

Frequency bands were defined primarily based on values used in the literature (e.g. Siegel et al *PNAS* 2009), and then the significance of the difference in both power and PPL for correct vs. wrong trials was evaluated for each band-- see Table S2 for statistics of power and PPL in all frequency bands. Only the beta band showed a significant increase in PPL with correct performance, which is why it is shown in Fig. 2b and focused on in further analysis. There is a decrease in gamma band PPL in correct trials.

Also, for the dependence on stimulus Pref for SPL difference, it's not clear that the number of spikes (which is of course different in these conditions) would not bias these estimates?

We thank the reviewer for prompting us to clarify this point. SPL was calculated using a sliding window with a fixed number of spikes, so that differences in firing rate should not affect the SPL value (see Methods). We have clarified this point in the Methods section: *"In order to control for any effects of different numbers of spikes and trials between the two conditions, we considered a fixed window of 15 spikes and measured the SPL magnitude for the spikes of each window, and then took the mean across these windows⁴⁷."*

The IT spiking rate is also not significantly different for correct vs. wrong trials during the delay period, when we see a difference in the SPL.

6. For the data shown in Figure 4, what's the relation between the sorting and performance (related to point 2, above).

Only correct trials are shown in the High/Low PPL analysis in this figure. Unfortunately there were not enough wrong trials for us to calculate ROC values between High and Low PPL.

7. More details about the eye movements made within the 1.5 dva should be provided, given that the data set relies so heavily on signals obtained from spatially selective FEF sites, where movements of the eyes on certain trials would certainly affect activity. Can, for example, raw eye position data along with stimulus location, predict behavior? We have added a figure and statistics evaluating eye position and microsaccades during the delay period to the Supplementary Information (Fig. S1e-f). Although there was a bias in eye position based on sample location, there was no difference in eye position for correct vs wrong trials ($\Delta X_{pos} = 0.024 \pm 0.012$, $p = 0.190$, $\Delta Y_{pos} = 0.013 \pm 0.009$, $p = 0.256$, $n = 86$; Fig. S1e). There was also no difference in microsaccade rate between correct and wrong trials ($\Delta \text{MicrosaccadeRate} = 0.079 \pm 0.068$, $p = 0.208$, $n = 86$; Fig. S1f).

Minor Comments

1. Figure 1b: it's not really clear what we learn from this bar plot . One thing that seems apparent is that behavior between the two animals was actually significantly different? The number of sessions should be included here (and in the text)

The main point of inclusion is to give an idea of the proportion of correct and wrong trials, since having sufficient error trials is important for analysis involving behavioral correlations. The performance of the animals is different, see response to point 1 of Reviewer 2 above; relatedly, we now color-code data from the two monkeys in all scatter plots and provide separate statistical analysis in each animal (Table S1). We have added data on the number of sessions to the figure legend and text (51 sessions in M1 and 41 sessions in M2), and additional information on behavioral performance in the Supplementary Information (Fig. S1).

2. Figure 1c and 1d: why is no different plot shown?

Since the differences between correct and wrong are not significant, there is not much to see in a plot of the difference. We have added scatter plots examining the difference in firing rate and LFP power for correct vs. wrong trials in figure S2.

3. How the experimenters decide on the size of the stimulus set to use in the DMS task? Would you predict a smaller or larger set would affect the results of the study?

Three sample objects per session is essentially a minimum for the DMS task (since with only two objects the animal could potentially employ an A vs. not-A strategy, rather than remembering both samples). We used the minimum number of objects possible to maximize the number of trials per condition (which is especially critical for these PPL/SPL types of analysis). We have clarified this rationale in the Methods, *"Only three objects were used per recording session in order to maximize the number of trials (and statistical power) for each condition."* A larger stimulus set would cut down on the number of trials per condition, presumably weakening our statistical power-- unless, for example, multiple objects driving strong IT responses were then grouped back together for analysis. It seems likely that the animal's performance would be lower with a larger number of stimuli in a single session, although we did not test this.

4. Figure 3 introduces the acronym SPL before the text describes spike-phase locking

We have fixed the order.

5. Figure legend for 4 refers to "histograms" where it seems like there is only one histogram and the last sentence ("Same as (e) but for the Out condition.") seems out of place?

We have edited the legend.

6. Check for misspelling of trial as "trail"

Fixed.

REVIEWER COMMENTS

Reviewer #1 (Remarks to the Author):

I thank the authors for doing their best to address my concerns on the original manuscript. After careful consideration, I remain fully supportive for the questions they ask in this paper and for the premise of the study, and most of the issues raised in my original review have been addressed. However, an important technical concern remains unanswered. Specifically, I had several concerns regarding the strength of the effects emerging from their pairwise analysis reported in Figures 2, 3, and 4, and the confidence in the results reported in the manuscript. The main problem is the limited number of pairs in each animal (and Reviewer 3 had similar concerns), which is exacerbated by the fact that many pairs share the same electrodes. Ideally, the study should have been conducted with multiple electrodes (in their rebuttal the authors mention Neuropixels, but there are other, more accessible, techniques already in place they could have used), which would have increased the pair count by at least one order of magnitude.

The authors did not directly address this issue with additional multi-electrode recordings, but chose instead to engage in discussions arguing that their effects are statistically significant. Their data may cross the P value for significance, however this still does not address issues regarding the robustness of their results. I see two options for moving forward: 1) they could record extra sessions using multi-electrode arrays to increase the total number of pairs and hence reassure us that the effects are robust (this option implies more effort), or 2) faster route - perform a multiple comparison analysis, for instance the Bonferroni correction, to disclose how many of their pairs cross the statistical significance threshold. To be convinced by their conclusions, I would expect the great majority of pairs to show significance after the multiple comparisons test.

Reviewer #2 (Remarks to the Author):

The revised manuscript addressed well most of the comments and questions raised by this reviewer. The authors provided additional details about the behavioral performance and neuronal activity for individual monkeys.

One minor comment. The rationale for limiting the analysis of delay activity to the preferred stimuli is not supported by the data recorded in IT cited by the authors. In fact, Fuster in his JNP 1990 paper writes: "Most of the units immediately and selectively responding to the sample failed to show a subsequent and selective activation during the delay. Conversely, a number of units increased firing selectively in that delay after having failed to do so in the sample period". Similar observations have been seen for other non-spatial tasks during prefrontal recordings (eg Hussar & Pasternak, 2012).

Otherwise, the revised manuscript is greatly improved. The study is novel, well designed and carefully

analyzed. It provides convincing documentation in support of the interactions between prefrontal neurons and neurons processing sensory signals in the retention and retrieval of these signals during working memory tasks.

Reviewer #3 (Remarks to the Author):

In this revised ms, the authors have adequately addressed my concerns. I think the general finding - that coupling between FEF and IT in the beta band, reflects performance, whereas spiking and LFP activity in each area alone is less clearly linked to behavior. Figure 3 and the associated results, suggesting that beta band FEF phase leads IT for correct trials but not wrong trials is also quite interesting, providing an important clue about how information flows through the system during a DMS task like that used in this study.

The addition of the separate data for the two animals is critical. They are not identical (RTs and RFs clearly differ), but the general pattern of results seems consistent. That said, the authors say little about the fairly large differences in very low frequency coupling predictive of behavior in M2 (Figure S3b, M2) and these data are not analyzed and included in Table S1? In Figure 2a, there is some residual effect, but clearly as this is only from M2, it's diluted.

Minor comments:

Figure 3: Fixation is misspelled

Figure S1: Mixture of behavior and spike sorting doesn't make sense

Figure legend S1:

Explain that the two markers are for M1 and M2 in a-d (add inset?)

August 5, 2020

Dear Reviewers,

We have revised our manuscript NCOMMS-20-02480-T, entitled “Frontotemporal Coordination Predicts Working Memory Performance and its Local Neural Signatures”, based on your latest comments. We thank you for your comments in this and the previous round, as addressing these issues has substantially improved the manuscript’s clarity and rigor. In accordance with these comments, we have added an analysis of selectivity during the different task periods, and a measure of effect size was added to table S1. Point by point responses to each reviewer’s concerns are included below.

Reviewer #1 (Remarks to the Author):

I thank the authors for doing their best to address my concerns on the original manuscript. After careful consideration, I remain fully supportive for the questions they ask in this paper and for the premise of the study, and most of the issues raised in my original review have been addressed.

However, an important technical concern remains unanswered. Specifically, I had several concerns regarding the strength of the effects emerging from their pairwise analysis reported in Figures 2, 3, and 4, and the confidence in the results reported in the manuscript. The main problem is the limited number of pairs in each animal (and Reviewer 3 had similar concerns), which is exacerbated by the fact that many pairs share the same electrodes. Ideally, the study should have been conducted with multiple electrodes (in their rebuttal the authors mention Neuropixels, but there are other, more accessible, techniques already in place they could have used), which would have increased the pair count by at least one order of magnitude.

We thank the reviewer for their support and appreciation for the significance of the study. However, we must take issue with the statement that “many pairs share the same electrodes” and that there are methods that “would have increased the pair count by at least one order of magnitude”. We would like to emphasize that figures 2-3 solely rely on LFP analysis; in each of these analyses, the LFP recording from each electrode is used only once. There are never multiple LFP pairs that share the same electrodes.

Indeed, for the LFP analysis, linear array recordings would not provide much benefit—LFP signals recorded from neighboring array contacts are highly correlated, and we don’t believe we can count them as independent signals for the purposes of LFP analysis. Therefore, we would like to draw the reviewer’s attention to the point that the single electrode recordings we performed were not inferior to array recordings for our main goal of LFP-LFP coordination (figures 2-3), nor would linear arrays have provided “an order of magnitude increase” in the pair count for LFP-LFP analysis, at least not such an increase

in the number of *independent* pairs. Thus, accessing IT with rigid single electrode through a guide tube remains our preferred method to assess LFP-LFP coordination.

For figures that include spiking activity (Fig 4 and Fig 5), it is true that array recordings could have boosted the number of IT spike-LFP pairs available for this analysis. Having chosen to use single electrodes, at the expense of lower yield of single unit recording, we then spent more time during the experimental stage to isolate more neurons per electrode. To address the reviewer's concerns about isolating multiple neurons on a single electrode, we assessed the isolation quality using an SVM analysis of waveforms, and also verified that the results remained when selecting just a single neuron from each electrode. We wish to note that the number of pairs we report is by no means outside the norm in the field for dual area recordings—for example, one FEF-IT paper reported a total of 24 neuron pairs (Monosov et al. PNAS 2010). Gregoriou et al. (Neuron 2012) break their larger dataset down by response properties and compare inter-area spike-LFP coherence measurements between groups with n of 58 and 49 neuron-LFP pairs, indicating that these measurements can be robust with pair numbers comparable to our own. Lastly, and perhaps most importantly, in response to the reviewer's concern, the robustness of all of our results is now directly quantified and reported in the revised paper (see below).

The authors did not directly address this issue with additional multi-electrode recordings, but chose instead to engage in discussions arguing that their effects are statistically significant. Their data may cross the P value for significance, however this still does not address issues regarding the robustness of their results. I see two options for moving forward: 1) they could record extra sessions using multi-electrode arrays to increase the total number of pairs and hence reassure us that the effects are robust (this option implies more effort), or 2) faster route - perform a multiple comparison analysis, for instance the Bonferroni correction, to disclose how many of their pairs cross the statistical significance threshold. To be convinced by their conclusions, I would expect the great majority of pairs to show significance after the multiple comparisons test.

We thank the reviewer for suggesting options to address the issue raised. Whether choosing option 1 or option 2, the ultimate aim is to determine if the results are robust enough to report. To accomplish this, we chose to assess robustness directly using a standard statistical measure of effect size, specifically Cohen's d. We have added a column to Table S1 reporting the Cohen's d value for all the major analyses of the paper. All major findings have a d value of >2, which is considered a "huge effect" (Sawilowsky *J Mod App Stat Meth* 2009). For the purpose of putting these values in the context of other published relevant work, here we provide a table summarizing the effect sizes of results recently reported in the Nature journal family, which range from ~0.5 to 5.0.

Source	Cohen's d	Link
Rideaux & Welchman Nat Comm 2018	0.74, 0.78, 0.86, 1.49, 2.54	https://www.nature.com/articles/s41467-018-03400-y
Dandolo & Schwabe Nat Comm 2018	0.54, -1.02, 1.94, -2.65, 4.18, 4.49	https://www.nature.com/articles/s41467-018-03661-7

Bartram et al. Nat Comm 2017	1.92, 1.98	https://www.nature.com/articles/s41467-017-00748-5
Tamaki et al. Nat Neuro 2020	0.67, 1.09, 1.13, 1.69, 1.90	https://www.nature.com/articles/s41593-020-0666-y
Amadei et al. Nature 2017	0.94	https://www.nature.com/articles/nature22381
Twining et al. Nat Neuro 2017	1.12, 1.26, 3.34, 5.25	https://www.nature.com/articles/nn.4481
Bein et al. Nat Comm 2020	0.55, 0.57, 0.72	https://www.nature.com/articles/s41467-020-17287-1
Yang et al. Nat Neuro 2020	0.479, 0.709, 1.142	https://www.nature.com/articles/s41593-020-0630-x
Tarazona et al. Nat Comm 2020	1.98	https://www.nature.com/articles/s41467-020-16937-8

As all of the reviewers likely know, Cohen's d provides a standardized measure of mean differences that allows for comparisons of effects across studies (i.e. meta-analyses), and thus we know explicitly that our results scale well with other noteworthy published results. This is in contrast to simply counting the number of individually significant observations (e.g. pairs), which in neurophysiological studies can be confounded by extraneous factors like the number of trials, or the time period of activity.

To summarize, we deliberately chose acute recordings with two single electrodes to address the main goal of studying LFP-LFP coordination between areas. This approach increased our ability to move the electrodes (compared to surface arrays), and it provided more rigidity to reach IT (compared to linear arrays, which would provide the same LFP yield). The disadvantage was the extra time and effort during the experimental stage to acquire sufficient data. Any potential concerns about the quality of isolations have been rigorously addressed by assessing the recording quality (SVM analysis), as well as subsampling and bootstrapping. Concerns about the robustness of the results were addressed directly by measuring the power of the analysis (Cohen's d). We would like to thank the reviewer for prompting us to add these analyses, which have substantially improved the paper.

Reviewer #2 (Remarks to the Author):

The revised manuscript addressed well most of the comments and questions raised by this reviewer. The authors provided additional details about the behavioral performance and neuronal activity for individual monkeys.

We thank the reviewer for this general assessment.

Minor comments.

One minor comment. The rationale for limiting the analysis of delay activity to the preferred stimuli is not supported by the data recorded in IT cited by the authors. In fact, Fuster in his JNP 1990 paper writes: "Most of the units immediately and selectively responding to the sample failed to show a subsequent and selective activation during the delay. Conversely, a number of units increased firing selectively in that delay after having failed to do so in the sample period". Similar observations have been seen for other non-spatial tasks during prefrontal recordings (eg Hussar & Pasternak, 2012).

The reviewer is correct that this paper was not a good choice of reference for this point; it is also true that the relationship between visual selectivity and delay period activity is more complicated than a simple perfect correlation- as the reviewer notes, many visually selective neurons show no delay activity, and other neurons with no visual activity respond during the delay. (In our case, neurons were screened initially for a visual response prior to beginning DMS recordings, eliminating the latter group.) Despite these complications, there have been previous reports of an overall correlation between visual and delay selectivity (Chelazzi et al. *J. Neurophys.* 1998).

Note that we defined the preferred object based not solely on the visual period, but based on the maximum response across the visual and delay periods (as in Hung et al. *Science* 2005). We have added an analysis in figure S2b, in which we examine the correlation in our own dataset between selectivity in the visual and delay periods- on average selectivity in these periods is positively correlated.

Otherwise, the revised manuscript is greatly improved. The study is novel, well designed and carefully analyzed. It provides convincing documentation in support of the interactions between prefrontal neurons and neurons processing sensory signals in the retention and retrieval of these signals during working memory tasks.

We thank the reviewer for this comment.

Reviewer #3 (Remarks to the Author):

In this revised ms, the authors have adequately addressed my concerns. I think the general finding - that coupling between FEF and IT in the beta band, reflects performance, whereas spiking and LFP activity in each area alone is less clearly linked to behavior. Figure 3 and the associated results, suggesting that beta band FEF phase leads IT for correct trials but not wrong trials is also quite interesting, providing an important clue about how information flows through the system during a DMS task like that used in this study.

We thank the reviewer for this comment.

The addition of the separate data for the two animals is critical. They are not identical (RTs and RFs clearly differ), but the general pattern of results seems consistent. That said, the authors say little about the fairly large differences in very low frequency coupling predictive of behavior in M2 (Figure S3b, M2) and these data are not analyzed and

included in Table S1? In Figure 2a, there is some residual effect, but clearly as this is only from M2, it's diluted.

We appreciate this comment. We followed the general practice of discussing results which are consistent across animals, and we report the combined statistics for all frequency bands in Table S2. However, we have now added theta band PPL results for both monkeys to Table S1. As the reviewer was able to tell from the figure, this effect is significant in M2 alone, but the trend is opposite in M1 and when combined there is no significant effect.

Minor comments:

Figure 3: Fixation is misspelled

Fixed, thanks.

Figure S1: Mixture of behavior and spike sorting doesn't make sense

Spike sorting results were moved into Figure S2 in the revised manuscript.

Figure legend S1:

Explain that the two markers are for M1 and M2 in a-d (add inset?)

We have placed this clarification at the start of the figure legend.

References:

- Amadei, E. A., Johnson, Z. V., Jun Kwon, Y., Shpiner, A. C., Saravanan, V., Mays, W. D., Ryan, S. J., Walum, H., RAINNIE, D. G., Young, L. J., & Liu, R. C. (2017). Dynamic corticostriatal activity biases social bonding in monogamous female prairie voles. *Nature*, *546*(7657), 297–301. <https://doi.org/10.1038/nature22381>
- Bartram, J., Kahn, M. C., Tuohy, S., Paulsen, O., Wilson, T., & Mann, E. O. (2017). Cortical Up states induce the selective weakening of subthreshold synaptic inputs. *Nature Communications*, *8*(1), 665. <https://doi.org/10.1038/s41467-017-00748-5>
- Bein, O., Duncan, K., & Davachi, L. (2020). Mnemonic prediction errors bias hippocampal states. *Nature Communications*, *11*(1), 3451. <https://doi.org/10.1038/s41467-020-17287-1>
- Chelazzi, L., Duncan, J., Miller, E. K., & Desimone, R. (1998). Responses of neurons in inferior temporal cortex during memory-guided visual search. *Journal of Neurophysiology*, *80*(6), 2918–40.
- Dandolo, L. C., & Schwabe, L. (2018). Time-dependent memory transformation along the hippocampal anterior–posterior axis. *Nature Communications*, *9*(1), 1205. <https://doi.org/10.1038/s41467-018-03661-7>
- Gregoriou, G. G., Gotts, S. J., & Desimone, R. (2012). Cell-type-specific synchronization of neural activity in FEF with V4 during attention. *Neuron*, *73*(3), 581–94.
- Hung, C. P., Kreiman, G., Poggio, T. A., & DiCarlo, J. J. (2005). Fast readout of object identity from macaque inferior temporal cortex. *Science*, *310*(5749), 863–6. <https://doi.org/10.1126/science.1117593>
- Monosov, I. E., Sheinberg, D. L., & Thompson, K. G. (2010). Paired neuron recordings in the prefrontal and inferotemporal cortices reveal that spatial selection precedes object identification during visual search. *Proceedings of the National Academy of Sciences*, *107*(29), 13105–10. <https://doi.org/10.1073/pnas.1002870107>
- Rideaux, R., & Welchman, A. E. (2018). Proscription supports robust perceptual integration by suppression in human visual cortex. *Nature Communications*, *9*(1), 1502. <https://doi.org/10.1038/s41467-018-03400-y>
- Sawilowsky, S. (2009). New Effect Size Rules of Thumb. *Journal of Modern Applied Statistical Methods*, *8*(2). <https://doi.org/10.22237/jmasm/1257035100>
- Tamaki, M., Wang, Z., Barnes-Diana, T., Guo, D., Berard, A. V., Walsh, E., Watanabe, T., & Sasaki, Y. (2020). Complementary contributions of non-REM and REM sleep to visual learning. *Nature Neuroscience*, 1–7. <https://doi.org/10.1038/s41593-020-0666-y>
- Tarazona, S., Balzano-Nogueira, L., Gómez-Cabrero, D., Schmidt, A., Imhof, A., Hankemeier, T., Tegnér, J., Westerhuis, J. A., & Conesa, A. (2020). Harmonization of quality metrics and power calculation in multi-omic studies. *Nature Communications*, *11*(1), 3092. <https://doi.org/10.1038/s41467-020-16937-8>
- Twining, R. C., Vantrease, J. E., Love, S., Padival, M., & Rosenkranz, J. A. (2017). An intra-amygdala circuit specifically regulates social fear learning. *Nature Neuroscience*, *20*(3), 459–469. <https://doi.org/10.1038/nn.4481>
- Yang, J., Zhang, H., Ni, J., De Dreu, C. K. W., & Ma, Y. (2020). Within-group synchronization in the prefrontal cortex associates with intergroup conflict. *Nature Neuroscience*, *23*(6), 754–760. <https://doi.org/10.1038/s41593-020-0630-x>

REVIEWER COMMENTS

Reviewer #1 (Remarks to the Author):

I appreciate the efforts made by the authors to clarify and respond to the issues raised in my previous rounds of review. As I pointed out in my last review, although I find some of the responses convincing, I am still unclear about how the significance of the effects (correct vs. wrong comparisons) is computed in each case. This is an important issue, as most of the results depend on that. Therefore, I urge the authors to provide a very detailed description of it in the Results section and the Methods.

My main concern is that the statistical significance of the results in Figures 2, 3, and 4 is unclear. The reason of my concern is that: 1) they only have a limited number of pairs in each animal (due to technical constraints), and some of the neurons 'share' the same electrode, and 2) there are multiple comparisons in the statistical tests between correct/wrong responses and multiple analyses on the same spike-spike and spike-LFP pairs. However, from my reading it seems that there is no control over these multiple comparisons and tests. I urged the authors to perform multiple comparison analyses, such as the Bonferroni correction, to disclose how many of their pairs exhibit statistically significant effects for the analyses in Figures 2, 3, and 4, but they haven't done so.

As I mentioned in my previous review, to be convinced by their conclusions I would expect a great majority of pairs to show statistical significance after the multiple comparisons test is performed. This is not difficult to do and not an unusual request at all.

Reviewer #2 (Remarks to the Author):

The latest revision and the rebuttal letter addressed well my remaining reservations. I also thought the points raised by the other two reviewer were also well addressed. I strongly recommend that the paper is published in Nature Communications.

Reviewer #3 (Remarks to the Author):

The revised version of the ms satisfies all of my concerns.

September 18, 2020

Dear Reviewers,

We have revised our manuscript NCOMMS-20-02480-T, entitled “Frontotemporal Coordination Predicts Working Memory Performance and its Local Neural Signatures”, based on your latest comments. We thank you for your comments in this and the previous rounds, as addressing these issues has substantially improved the manuscript’s clarity and rigor. In accordance with suggestions made by the reviewers, we have added plots to supplementary figure S2 for evaluating the quality of spike sorting and its impact on the results involving spiking activity. Point by point responses to each reviewer’s concerns are included below.

Reviewer #2 (Remarks to the Author):

The latest revision and the rebuttal letter addressed well my remaining reservations. I also thought the points raised by the other two reviewer were also well addressed. I strongly recommend that the paper is published in Nature Communications.

We appreciate the reviewer saying that we have addressed the other reviewer’s points well in addition to their own concerns.

Reviewer #3 (Remarks to the Author):

The revised version of the ms satisfies all of my concerns.

We thank the reviewers for their helpful suggestions and support.

Reviewer 1 (Remarks to the Author):

I appreciate the efforts made by the authors to clarify and respond to the issues raised in my previous rounds of review. As I pointed out in my last review, although I find some of the responses convincing, I am still unclear about how the significance of the effects (correct vs. wrong comparisons) is computed in each case. This is an important issue, as most of the results depend on that. Therefore, I urge the authors to provide a very detailed description of it in the Results section and the Methods.

All statistical comparisons are performed at the level of the population. The relevant measure (firing rate, LFP power, PPL, etc.) is computed for each neuron (or LFP, or LFP-LFP pair) for Correct trials and for Wrong trials. A paired comparison of the values of Correct vs. Wrong is then made across the population using the Wilcoxon signed-rank test. We have clarified this procedure in the Methods section on statistical analysis, which now reads

“Statistical analyses were performed using the Wilcoxon’s signed-rank test (for paired comparisons) or rank sum (for unpaired comparisons), unless otherwise specified. Statistical comparisons were performed across the population. For example, in comparing PPL for correct vs. wrong trials: for each LFP-LFP pair, PPL in each frequency band was calculated for correct trials, and for wrong trials. The Wilcoxon sign-rank test was then used to compare PPL values for correct trials vs. wrong trials across the population. Statistics for individual monkeys are reported in Table S1, and behavioral correlation statistics for all frequency bands and metrics are reported in Table S2.”

My main concern is that the statistical significance of the results in Figures 2, 3, and 4 is unclear. The reason of my concern is that: 1) they only have a limited number of pairs in each animal (due to technical constraints), and some of the neurons 'share' the same electrode, and 2) there are multiple comparisons in the statistical tests between correct/wrong responses and multiple analyses on the same spike-spike and spike-LFP pairs. However, from my reading it seems that there is no control over these multiple comparisons and tests. I urged the authors to perform multiple comparison analyses, such as the Bonferroni correction, to disclose how many of their pairs exhibit statistically significant effects for the analyses in Figures 2, 3, and 4, but they haven't done so.

As I mentioned in my previous review, to be convinced by their conclusions I would expect a great majority of pairs to show statistical significance after the multiple comparisons test is performed. This is not difficult to do and not an unusual request at all.

In this revision, we further examined whether units from the same electrode showed a correlation in their effect size (for the SPL effect in 4a and the MI effect in 5e). There was no correlation in effect size for neurons recorded on the same electrode (Fig. S2d), nor was the correlation in effect any different for same-electrode pairs compared to different-electrode pairs (Fig. S2e&f).

We emphasize once again that for the analysis in figures 2 and 3, which involve LFP signals but not spiking activity, no recording contributes more than once. To address the reviewer's concerns about isolating multiple neurons on a single electrode, we assessed the isolation quality using an SVM analysis of waveforms in Fig S2a. We also verified that the results are valid even when selecting just a single neuron from each electrode (Fig S2b&c), indicating that using multiple units from the same electrode does not drive the effect.

All of these controls are now presented together in Fig. S2 and the accompanying text

“Spike sorting isolation quality:

We cross validated the spike sorting quality using an SVM Classifier. Figure S2a shows the average classifier performance in categorizing the spike waveforms of simultaneously recorded neurons across all sessions. The overall high performance of the classifier indicates that the spikes of each sorted cluster are well isolated from each other cluster ($\Delta Perf_{IT} = 98.80\% \pm 2.50$, $n = 301$ pairs, $p < 10^{-50}$, $\Delta Perf_{FEF} = 98.67\% \pm 4.018$, $n = 105$ pairs, $p < 10^{-18}$ compared to chance level, i.e. 50%).

To doubly control for the possibility of oversorting, we verified results involving spiking activity using only a single randomly selected spiking unit from each recording session (Fig. S2b). For the difference in SPL for correct vs. wrong trials (Fig. 4a), we measured the distribution of the median difference in correct vs. wrong SPL for 1000 random subsamples of 79 spike-LFP pairs, each with a maximum of one unit per recording session. For > 99.5% of the subsamples the SPL for correct was greater than wrong, and the median of this distribution was significantly greater than zero ($\Delta\text{SPL}_{\text{Cr-Wr}} = 0.008 \pm 0.000$, $p < 10^{-10}$, $n = 1000$). For the modulation of object selectivity on High vs. Low PPL trials (Fig. 5e), we calculated the distribution of the median difference in modulation index (modulation of object discriminability for High vs. Low PPL trials) for In vs. Out, for 1000 subsamples (each with 40 units, a maximum of one unit per recording session, Fig. S2c). For 100% of the subsamples the modulation index was greater for In vs. Out trials, and the median of this distribution was significantly greater than zero ($\Delta\text{MI}_{\text{In-Out}} = 0.090 \pm 0.000$, $p < 10^{-10}$, $n = 1000$). Thus the significance of the SPL and MI results did not depend on having multiple spiking units from the same electrode included in the analysis. Additionally, we tested whether neurons recorded from the same electrode showed a correlation in their SPL and MI effects. We calculated the correlation between the effect reported in figure 4a (SPL correct – wrong) for pairs of neurons recorded on the same electrode, and found no correlation (Fig. S2d; $r = 0.033$, $p = 0.210$). Similarly for the MI effect from figure 5e (MI In-Out), there was no correlation in effect size between neurons recorded on the same electrode ($r=0.1$, $p=0.110$). We further compared the magnitude of these correlation values with the distribution of correlations for randomly selected pairs of neurons recorded on different electrodes (Fig. S2e-f); there was no significant difference between the correlation observed for same-electrode pairs and the median of the distribution of correlations for different-electrode pairs (correlation for same vs. different electrode pairs: SPL, $p = 0.160$; MI, $p = 0.100$). Altogether, these analyses confirm that sorting multiple spiking units from single electrode recordings did not artificially inflate the magnitude or significance of the reported effects.”

To summarize, any potential concerns about over-sampling by using multiple units from the same electrode have been rigorously addressed by assessing the isolation quality (SVM analysis), verifying that the results persist even when using only one unit per recording session, and demonstrating that the effects are not even correlated for units recorded on the same electrode. Concerns about the robustness of the results were addressed directly by measuring the power of the analysis (Cohen’s d). We thank the reviewer for prompting us to add these analyses, which have substantially improved the paper.

REVIEWERS' COMMENTS

Reviewer #1 (Remarks to the Author):

The manuscript continues to be improved. However, the authors did not perform a crucial statistical analysis that I requested in my previous round of review, i.e., the multiple comparisons analysis, such as the Bonferroni correction. This is the only way to ensure that their results are robust and statistically sound, for the reasons laid out in my previous review but reiterated below.

This analysis is needed since they are literally performing multiple comparisons using the same neural data originating from multiple recording sites (mainly in relation to the LFP data and associated measures). For instance, they are comparing different measures between correct and wrong responses in FEF and IT, such as firing rates, LFP power in multiple frequency bands (theta, alpha, beta, low gamma, high gamma), phase-phase locking (PPL) in each frequency band, object selectivity of PPL, inter-area spike-phase locking (SPL) in each frequency band, etc., and some of these measures are examined both within each area and across areas. Since these are multiple comparisons simultaneously performed on the data originating from the same recording sites, it is likely that some of these comparisons will turn statistically significant simply by chance if the alpha value is held fixed at 0.05.

Although I urged them to perform this analysis in my previous review, the authors don't even mention it in their rebuttal, and this is a significant source of concern.

Reviewer #2 (Remarks to the Author):

I believe that the revision addressed the remaining issues raised by the last review. The revised description of statistical analyses provided the needed clarification. The authors also carefully addressed the question of their spike sorting approach to isolating multiple neurons on a single electrode and statistical treatment of controls. The plots and the stats presented in Fig S2 effectively summarize various controls. They are convincing.

Reviewer #3 (Remarks to the Author):

The authors further analysis of the data obtained from the same electrode (shown in parts of Figures 4 and 5) address my remaining concerns for this ms.

November 10, 2020

Dear Reviewers,

We have revised our manuscript NCOMMS-20-02480-T, entitled “Frontotemporal Coordination Predicts Working Memory Performance and its Local Neural Signatures”, based on your latest comments. We thank you for your comments in this and the previous rounds, as addressing these issues has substantially improved the manuscript’s clarity and rigor. Point by point responses to each reviewer’s concerns are included below.

Reviewer #2 (Remarks to the Author):

I believe that the revision addressed the remaining issues raised by the last review. The revised description of statistical analyses provided the needed clarification. The authors also carefully addressed the question of their spike sorting approach to isolating multiple neurons on a single electrode and statistical treatment of controls. The plots and the stats presented in Fig S2 effectively summarize various controls. They are convincing.

We appreciate the reviewer saying that we have addressed the other reviewer’s points well in addition to their own concerns.

Reviewer #3 (Remarks to the Author):

The authors further analysis of the data obtained from the same electrode (shown in parts of Figures 4 and 5) address my remaining concerns for this ms.

We thank the reviewer for their helpful suggestions and support.

Reviewer 1 (Remarks to the Author):

The manuscript continues to be improved. However, the authors did not perform a crucial statistical analysis that I requested in my previous round of review, i.e., the multiple comparisons analysis, such as the Bonferroni correction. This is the only way to ensure that their results are robust and statistically sound, for the reasons laid out in my previous review but reiterated below. This analysis is needed since they are literally performing multiple comparisons using the same neural data originating from multiple recording sites (mainly in relation to the LFP data and associated measures). For instance, they are comparing different measures between correct and wrong responses in FEF and IT, such as firing rates, LFP power in multiple frequency bands (theta, alpha, beta, low gamma, high gamma), phase-phase locking (PPL) in each frequency band, object selectivity of PPL, inter-area spike-phase locking (SPL) in each frequency band, etc., and some of these measures are examined both within each area and across areas. Since these are multiple comparisons simultaneously performed on the data originating from the same recording sites, it is likely that some of these comparisons will turn statistically significant simply by chance if the alpha value is held fixed at 0.05. Although I urged them to perform this analysis in my previous review, the authors don’t even mention it in their rebuttal, and this is a significant source of concern.

We have revised the paper to address the multiple comparison concern. Although the robustness of the results was already addressed to the satisfaction of the other two reviewers by the addition of isolation quality and effect size analyses, and although performing additional, multiple comparison, tests on different measures (e.g., different frequency bands, SPL, PPL, etc) is neither appropriate nor common in the field¹⁻¹¹, we have nonetheless revised the paper to indicate that our key results are robust enough to survive multiple comparisons. Our key findings show significance of $p < 0.001$, and thus would still be significant with multiple comparison tests (up to $n=50$), if one assumes such a correction is needed (Table S1 caption). This is now stated in the main text (Discussion section, page 12):

“The coordination between the FEF and IT occurred mostly within the beta band, and the consistency of modulations in only this band across multiple measures (e.g. SPL, PPL) demonstrates the robust involvement of this frequency band. Moreover, given their level of significance, the beta band effects remain significant even when subjected to multiple comparison tests (e.g. Bonferroni’s correction, see Table S1).”

1. Lundqvist, Mikael, et al. "Gamma and beta bursts during working memory readout suggest roles in its volitional control." *Nature communications* 9.1 (2018): 394.
2. Brincat, Scott L., and Earl K. Miller. "Frequency-specific hippocampal-prefrontal interactions during associative learning." *Nature neuroscience* 18.4 (2015): 576-581.
3. Kayser, Christoph, et al. "Spike-phase coding boosts and stabilizes information carried by spatial and temporal spike patterns." *Neuron* 61.4 (2009): 597-608.
4. Belitski, Andrei, et al. "Low-frequency local field potentials and spikes in primary visual cortex convey independent visual information." *Journal of Neuroscience* 28.22 (2008): 5696-5709.
5. Montemurro, Marcelo A., et al. "Phase-of-firing coding of natural visual stimuli in primary visual cortex." *Current biology* 18.5 (2008): 375-380.
6. Fries, Pascal, et al. "Modulation of oscillatory neuronal synchronization by selective visual attention." *Science* 291.5508 (2001): 1560-1563.
7. Guillem, Karine, and Serge H. Ahmed. "Reorganization of theta phase-locking in the orbitofrontal cortex drives cocaine choice under the influence." *Scientific Reports* 10.1 (2020): 1-12.
8. Navas-Olive, Andrea, et al. "Multimodal determinants of phase-locked dynamics across deep-superficial hippocampal sublayers during theta oscillations." *Nature Communications* 11.1 (2020): 1-14.
9. Duvarci, Sevil, et al. "Impaired recruitment of dopamine neurons during working memory in mice with striatal D2 receptor overexpression." *Nature communications* 9.1 (2018): 1-13.
10. Tort, Adriano BL, et al. "Cortical networks produce three distinct 7–12 Hz rhythms during single sensory responses in the awake rat." *Journal of Neuroscience* 30.12 (2010): 4315-4324.
11. Salazar, R. F., et al. "Content-specific fronto-parietal synchronization during visual working memory." *Science* 338.6110 (2012): 1097-1100.